# MCPMARK: A BENCHMARK FOR STRESS-TESTING REALISTIC AND COMPREHENSIVE MCP USE

**Zijian Wu**[*][1,4], **Xiangyan Liu**[*][1,4], **Xinyuan Zhang**[* 2], **Lingjun Chen**[* 1], **Fanqing Meng**[* 1,4],
**Lingxiao Du**[* 1,4], **Yiran Zhao**[* 1], **Fanshi Zhang**[* 2,3], **Yaoqi Ye**[1], **Jiawei Wang**[2], **Zirui Wang**[2],
**Jinjie Ni**[1], **Yufan Yang**[2,3], **Arvin Xu**[‡ 2,3], **Michael Qizhe Shieh**[‡ 1,4]

[1] National University of Singapore    [2] EvalSys    [3] LobeHub    [4] absolute AI

## ABSTRACT

The MCP standardizes how LLMs interact with external systems, forming the foundation for general agents. However, existing MCP benchmarks remain narrow in scope: they focus on read-heavy tasks or tasks with limited interaction depth, and fail to capture the complexity and realism of real-world workflows. To address this gap, we propose `MCPMark`, a benchmark designed to evaluate MCP use in a more realistic and comprehensive manner. It consists of 127 high-quality tasks collaboratively created by domain experts and AI agents, each with a curated initial state and programmatic verification script. These tasks demand diverse CRUD operations and richer environmental interactions. We evaluate cutting-edge LLMs using a minimal agent framework. The best-performing model, `gpt-5-medium`, reaches only 52.56% pass@1 and 33.86% pass^4, while other strong models including `claude-sonnet-4` and `o3` fall below 30% pass@1 and 15% pass^4. On average, LLMs require 16.2 turns and 17.4 tool calls per task, highlighting the stress-testing nature of `MCPMark`.

**mcpmark.ai**    **eval-sys/mcpmark**

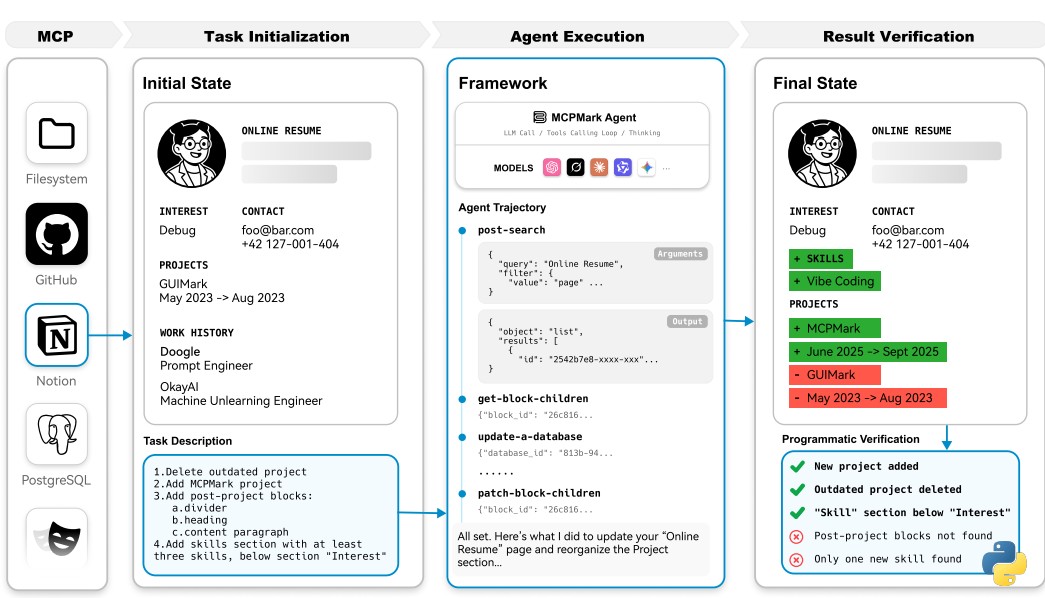

Figure 1: **`MCPMark` evaluation pipeline with full state tracking.** Each task begins from a curated initial state with a specific task instruction. The `MCPMark-Agent` then executes a tool-calling loop, followed by a programmatic verifier that evaluates whether all required checks are satisfied.

---

[*] Student leads; listed in random order.    [*] Equal contribution.    [‡] Equal advising.
Correspond to: {zijian.wu, liu.xiangyan}@u.nus.edu

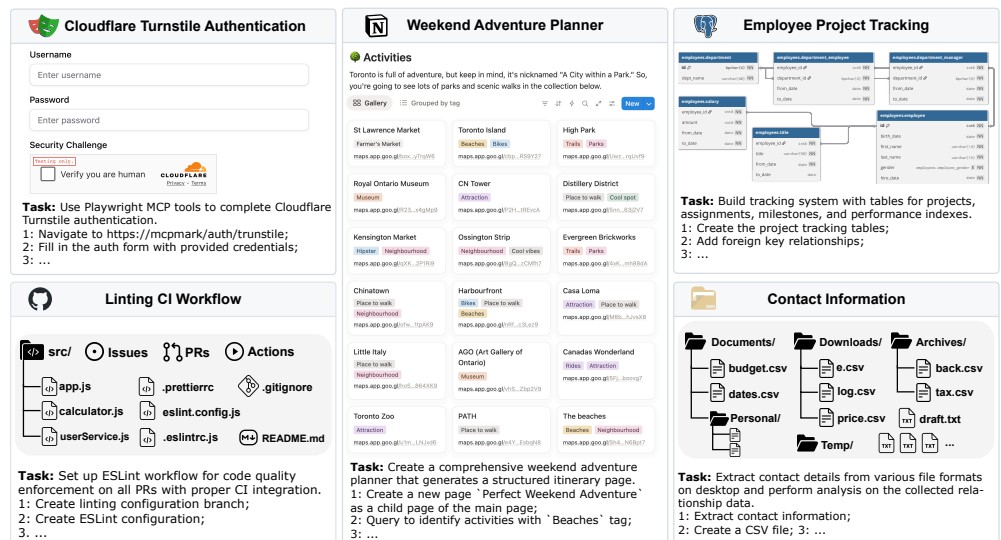

Figure 2: Representative task instances, showing initial states (**Top**) and task instructions (**Bottom**). Examples include: Login with Cloudflare Turnstile in *Playwright*; CI/CD setup with ESLint in *GitHub*; weekend planner using tagged queries in *Notion*; schema design for project tracking in *PostgreSQL*; and contact extraction to CSV in *Filesystem*. All tasks show complex, multi-step workflows typical of real-world usage.

# 1 INTRODUCTION

The Model Context Protocol (MCP) (Anthropic, 2024) is a standardized interface that connects large language models (LLMs) (Comanici et al., 2025; OpenAI, 2025c; Team, 2025) with external systems such as tools, APIs, databases, and contextual resources. By standardizing the way LLMs access and operate on these systems, MCP allows agents to function more effectively with "eyes and hands" in real environments, and many see it as a foundational layer for AI in the agentic era (Hou et al., 2025).

Despite growing use in practice, existing MCP benchmarks remain limited: tasks often involve shallow or read-heavy interactions (Liu et al., 2025; Yin et al., 2025; Mo et al., 2025; Luo et al., 2025), leading to a narrow range of task patterns. As a result, they fail to capture the complex, multi-step workflows typical of real-world usage. This makes it difficult to probe the performance boundaries—especially in assessing whether current models and agents possess the necessary capabilities, such as reasoning, planning, long-context processing, and tool use, to tackle realistic and demanding agent tasks.

To address these gaps, we introduce `MCPMark`, a benchmark designed to simulate realistic user scenarios within mirrored or isolated container environments, accompanied by reliable automated evaluation. Specifically, `MCPMark` spans five representative MCP environments: *Notion*, *GitHub*, *Filesystem*, *PostgreSQL* and *Playwright*. Figure 1 presents an overview of the evaluation pipeline, where each task comprises three components: task instruction, initial state, and programmatic verification script. Figure 2 provides examples of task instructions and initial states.

For task creation, after selecting or designing an initial state, a task instruction and verification script are developed through a *human–AI collaborative pipeline*, where domain experts and language agents iteratively co-design and refine each task. After this pipeline, we apply expert cross-review and community-level validation to ensure clarity, realism, and quality. Compared to existing MCP benchmarks, `MCPMark` offers significantly broader coverage of create, read, update, and delete (CRUD) operations across diverse workflows. In total, `MCPMark` comprises a total of 127 tasks and 38 unique initial states, with 20 to 30 tasks in each MCP environment.

To fairly evaluate model performance on these tasks, we introduce `MCPMark-Agent`, a minimal and general agent framework that executes models through a standardized tool-calling loop. `MCPMark-Agent` integrates with a variety of MCP servers and model providers, enabling

consistent and automated evaluation grounded in the programmatic infrastructure defined by `MCPMark`. Comprehensive experiments on state-of-the-art models demonstrate the benchmark's difficulty. Specifically, the best-performing model, `gpt-5-medium` (OpenAI, 2025c), achieves only 52.56% pass@1 and 33.86% pass^4, while other leading models such as `claude-sonnet-4` (Anthropic, 2025a) and `o3` (OpenAI, 2025d) fall below 30%

Table 1: Benchmark Comparison.

| Benchmark | Task Pattern | Verification | Average Turns |
|---|---|---|---|
| MCPEval | Synthetic | Hybrid | N/A |
| LiveMCPBench | CRUD-diverse | LLM-as-judge | 3.2 |
| MCP-Universe | Read-heavy | Programmatic | 6.8 |
| LiveMCP-101 | N/A | LLM-as-judge | 5.4 |
| MCPMark | CRUD-diverse | Programmatic | 16.2 |

pass@1 and 15% pass^4. On average, each task requires 16.2 execution turns and 17.4 tool calls, with some models such as `kimi-k2-instruct` (Team et al., 2025) averaging over 20 turns per task. Overall, as shown in Table 1, prior MCP benchmarks are limited by task patterns or verification rigor. In contrast, `MCPMark` integrates CRUD-diverse tasks, programmatic verification, and longer workflows, offering a closer alignment with real-world MCP use and workflow complexity.

In addition, our evaluation reveals several consistent patterns that underscore the distinctive properties of the benchmark. First, the benchmark demonstrates its **intrinsic difficulty through consistently low performance on the pass^4**, which more convincingly reflects real-world conditions than commonly used metrics like pass@1 or pass@4 (Yao et al., 2024), emphasizing the challenge of solving tasks reliably and consistently across multiple runs. Second, **performance varies substantially across different MCP environments**, suggesting a notable environment gap. This variation arises from differences in data availability and simulation fidelity: tasks involving local services such as the Filesystem are generally easier to emulate and more commonly represented in training data, whereas remote services like Notion require more complex, underrepresented interaction patterns that are harder to reproduce. Finally, the benchmark emphasizes **efficient tool use**: successful completions tend to involve fewer, more targeted tool calls, while failure cases often exhibit repetitive or exploratory interactions that fail to make meaningful progress. Collectively, these patterns show that `MCPMark` effectively surfaces key challenges in stability, generalization, and planning across diverse multi-component environments.

## 2 MCPMARK: STRESS-TESTING COMPREHENSIVE MCP USE

In this section, we provide a detailed introduction to `MCPMark`, including the benchmark construction process, the associated evaluation framework, and an overview of the benchmark.

### 2.1 BENCHMARK CONSTRUCTION

**MCP services and initial states.** `MCPMark` integrates 5 MCP servers that span diverse and practical application environments. A partial overview of each MCP tool set is shown in Figure 3 (right). Moreover, unlike prior work that uses generic or minimally initialized environments as task starting states (Liu et al., 2025; Luo et al., 2025; Yin et al., 2025), we carefully design initial states that reflect realistic and comprehensive usage scenarios, serving as the starting points for the tasks. Specifically:

- **Notion** connects to the official remote API for creating, editing, and querying both documents and databases. Initial states are instantiated from widely adopted templates.
- **GitHub** leverages the official remote API to support project management and Git operations, including CI/CD, issues, branches, pull requests, and commits. Initial states are derived from repositories with realistic development histories and configurations.
- **Filesystem** supports file I/O, directory organization, metadata inspection, and search. Initial states are curated folder structures that mirror everyday user scenarios.
- **PostgreSQL** provides access to a relational database, with tools for schema exploration and SQL query execution. Initial states are representative template databases with realistic schemas.
- **Playwright** enables browser automation, offering commands for navigation, form completion, data extraction, and generating screenshots or PDF exports. Initial states come from two sources: self-authored webpages designed to test specific functionalities (e.g., login through Cloudflare) and localhost webpages adapted from WebArena (Zhou et al., 2023).

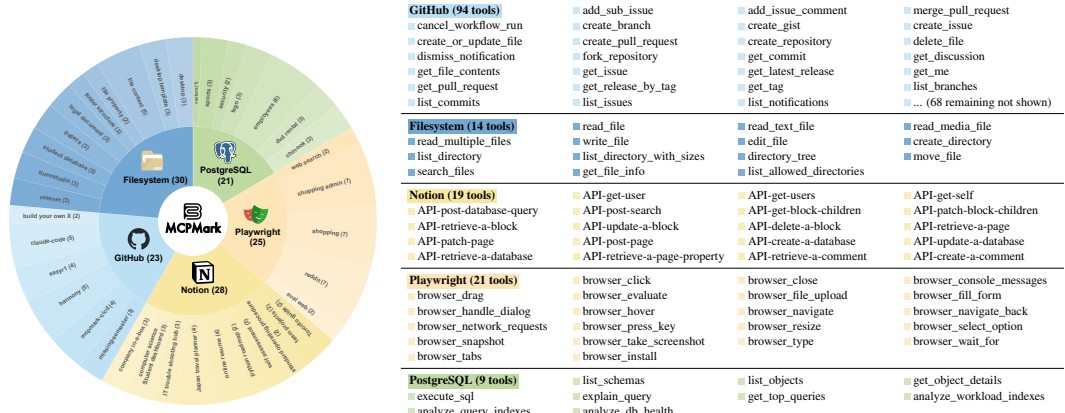

Figure 3: **Task distribution and tool set overview of `MCPMark`.** Left: 127 tasks distributed across 5 MCP servers and 38 curated initial states. Right: toolset per server, covering commonly used functionalities, with full support for CRUD operations in each corresponding MCP environment.

**Task creation pipeline.** Each task in `MCPMark` is grounded in an *initial state* of the respective environment (e.g., a template Notion page or a designated website) and consists of a *natural language instruction* paired with an *automatic verification script*. Constructing tasks of this form is difficult if we rely solely on humans or solely on agents. To address this, we design a human–AI collaborative pipeline that pairs human experts with two agents: a task creation agent and a task execution agent. The pipeline proceeds in four steps:

I. **Exploration**: Given an initial environment state, a human expert and the task creation agent jointly explore the environment, guided by a high-level instruction or topic informed by expertise and real-world experience. This stage aims to capture both a wide overview of the environment and deep, specific context that will later support realistic and well-grounded task creation.

II. **Evolvement**: The task creation agent proposes a new task instruction or refines an existing one by introducing additional complexity. This may include removing unnecessary instructions, increasing the difficulty of information seeking, raising the processing burden (e.g., through longer input content), or requiring more interaction steps. The human expert ensures that the task remains practical, verifiable, and sufficiently challenging.

III. **Verification**: The task creation agent drafts a programmatic verification script. The human expert then completes the task with assistance from the task execution agent. Afterward, the verification script is executed and iteratively refined until it is fully consistent with the task instruction. To ensure reliability, the human expert also adjusts the final environment state to validate whether the script correctly detects both successful and unsuccessful outcomes.

IV. **Iteration**: Steps ② and ③ are repeated to progressively increase task difficulty, while preserving automatic verifiability and maintaining realism through authentic user scenarios.

Overall, even with agent assistance, constructing each sample remains labor-intensive. Involving 10 experts with diverse backgrounds—including computer science PhD students, front-end designers, full-stack & AI infra engineers, and AI investors—each task takes $3 \sim 5$ hours of focused expert effort. While most tasks are built through the standard pipeline, experts occasionally leverage their accumulated experience or domain knowledge to directly write natural language instructions. In these cases, the task creation agent is bypassed, but the verification scripts are still generated and refined within the same pipeline. We defer the prompts and guidelines used in the task creation pipeline to Appendix B.

**Quality control.** All tasks underwent cross-review by human experts and a month-long community check to ensure clarity, consistency, and alignment with real-world application scenarios. In particular, for tasks that no model solved correctly, we conducted additional verification to ensure their validity. This process ensures that the benchmark remains challenging yet practical, and that evaluation outcomes are unambiguous.

## 2.2 BENCHMARK OVERVIEW

**Dataset statistics.** We create a total of 127 tasks across 5 MCP servers—30 for Filesystem, 28 for Notion, 25 for Playwright, 23 for GitHub, and 21 for PostgreSQL—based on 38 curated initial states. On average, the task instructions contain 288.6 words, and the corresponding verification scripts consist of 209.8 lines of code. The detailed task distribution is presented in Figure 3 (left), while the corresponding toolsets for each MCP are shown in Figure 3 (right).

**Task characteristics.** The tasks span a wide range of realistic workflows, including updating nested properties in Notion, managing commits and pull requests in GitHub, automating interactive forms in Playwright, organizing complex directory structures in the Filesystem, and executing transactional updates in PostgreSQL. Five representative tasks, one from each MCP, are shown in Figure 2. Collectively, these tasks provide diverse CRUD coverage and reflect the challenges of authentic multi-step workflows across varied application scenarios.

**Necessity of MCPMark.** Existing tool-use agent benchmarks such as AppWorld (Trivedi et al., 2024), WebArena (Zhou et al., 2023), and SWE-Bench (Jimenez et al., 2023) provide valuable testbeds, but they introduce trade-offs that limit their suitability for evaluating multi-step agent workflows in real-world use. SWE-Bench offers strong realism but is restricted to a single domain, whereas AppWorld and similar environments support diverse tasks but rely on custom wrappers or research-oriented APIs that do not reflect the stateful behavior of production systems.

In contrast, `MCPMark` builds directly on official MCP servers and real environments, enabling agents to interact with the same APIs and conditions they encounter in deployed settings. This design allows us to capture workflow complexity that cannot be reproduced in simulated environments, including CI/CD operations on live repositories, transactional database behavior, and multi-file organization in realistic file systems. As a result, `MCPMark` complements existing tool-use benchmarks by providing a high-fidelity evaluation setting tailored to modern MCP-based agent workflows.

## 2.3 EVALUATION FRAMEWORK

**State tracking and management.** `MCPMark` executes all tasks within sandboxed environments that enforce explicit state tracking, a design choice that ensures safety, reproducibility, and fair comparison across models. Each evaluation follows a consistent lifecycle: ① tasks begin from a well-defined initial state that mirrors realistic application scenarios, ② proceed with agent execution based on task instructions, and ③ conclude with an automatic verification script that programmatically checks whether the final environment satisfies the task requirements. After verification, ④ the environment is reset to its original state, preventing side effects and enabling repeated evaluation under identical conditions.

**Evaluation Agent.** To standardize evaluation, we provide `MCPMark-Agent`, a lightweight and general-purpose agent framework. It is built on LiteLLM together with the Model Context Protocol Python SDK to support compatibility and extensibility. Specifically, MCP servers are configured through the SDK, and their tools are exposed to the agent. LiteLLM then (1) converts the tools into the OpenAI function-call format and (2) routes requests to the official APIs of different providers, thereby ensuring execution that reflects each model's native capabilities.

During task evaluation, the agent follows a tool-calling loop in which the model iteratively invokes MCP tools, interprets responses from MCP servers, and adjusts its actions. The loop terminates once the model produces a final response *without further tool calls*. Although this agent framework is deliberately basic and omits optimizations that may be desirable in production systems (which we leave for future work), this design **avoids task-specific heuristics and model-specific biases**, thereby providing a clearer measure of a model's intrinsic agentic capabilities in MCP environments.

## 3 EXPERIMENTS

In this section, we describe the experimental setup, introduce the evaluated models and metrics, and present results and analyses on different environment, reasoning efforts, and failure patterns.

---

We also provide a subset of `MCPMark` comprising 50 tasks (10 per MCP server), derived from the standard tasks by relaxing subtask requirements or providing hints. The results are reported in Table 14.

https://github.com/BerriAI/litellm
https://github.com/modelcontextprotocol/python-sdk

## 3.1 Experimental Setup

**Models.** We test a range of state-of-the-art proprietary and open-source models, primarily accessed through LiteLLM. Proprietary models include `gpt-5` (OpenAI, 2025c) with different reasoning effort levels (`low`, `medium`, `high`) and smaller variants (`mini` and `nano`), as well as earlier `gpt-4.1` (OpenAI, 2025b) variants. We also evaluate `claude-opus-4.1`, `claude-sonnet-4`, `grok-4`, `grok-code-fast-1`, `o3`, `o4-mini`, `qwen3-max`, `gemini-2.5-flash`, and `gemini-2.5-pro` (Anthropic, 2025b;a; xAI, 2025; OpenAI, 2025d; Comanici et al., 2025). On the open-source side, we evaluate `qwen3-coder-plus`, `kimi-k2-instruct`, `deepseek-v3.1`, `glm-4.5`, and `gpt-oss-120b` (Team, 2025; Team et al., 2025; Liu et al., 2024; Zai, 2025; OpenAI, 2025a). We do not test small open-source models ($\leq$ 100B) due to the difficulty of the benchmark.

**Metrics.** We use three complementary metrics to measure agent performance: pass@1, pass@4, and pass^4. Pass@1, captures the single-run success rate, i.e., the proportion of tasks successfully in one single attempt. Pass@4 measures success when allowing up to 4 independent runs, indicating whether repeated attempts improve coverage of difficult cases. Pass^4 is a stricter measure: a task is counted as correct only if all four independent runs succeed, making it a strong indicator of model consistency and stability under stochastic generation (Yao et al., 2024).

**Implementation Details.** We use `MCPMark-Agent` as the unified framework to benchmark MCP use across models. While specialized agent designs could further improve performance, we leave such optimizations as important future work. Each run is limited to a maximum of 100 turns with a 3600-second timeout. Unless otherwise specified, all models are evaluated under their default inference settings (e.g., temperature, top-$p$, reasoning effort). The agent supports two execution paths: a general path via LiteLLM with function-calling tools and a native path with direct tool support for certain models (e.g., Anthropic API for extended thinking mode). For MCP server selection, we generally choose the most commonly used ones (see Appendix C for details).

## 3.2 Main Results

We evaluate all 127 tasks using `MCPMark-Agent`, reporting pass@1, pass@4, and pass^4 metrics. Unless otherwise specified, pass@1 scores are averaged over four independent runs and reported as mean $\pm$ std. Detailed results on each MCP service are provided in Appendix D, and representative trajectories appear in Appendix E.

**`MCPMark` remains challenging for frontier models.** Table 2 shows that the best-performing model, `gpt-5-medium`, reaches only 52.56% pass@1, while `qwen3-coder-plus`, the strongest open-source model, achieves 24.80%. Most proprietary models fall within the 15% to 30% range on pass@1, and several open-source models perform below 10%. Moreover, Table 9 highlights the high interaction demands of the benchmark: for example, `qwen3-max` and `kimi-k2-instruct` average 23.85/26.95 turns with 23.02/26.22 tool calls, respectively. These results underscore that `MCPMark` remains a highly challenging benchmark for current frontier models.

**Models generally perform better on local service tasks.** We observe from Table 2 that performance varies significantly across MCP services, showing a clear divide between local and remote environments. Local services such as PostgreSQL, Filesystem, and Playwright achieve substantially higher success rates, with `gpt-5-medium` reaching 76.19%, 57.50%, and 43.00% pass@1 respectively. Remote services like Notion and GitHub remain challenging, with most models achieving below 25% pass@1. This gap likely stems from data availability: local services are easier to simulate and collect training data for, while remote service APIs require authentic interaction traces that are expensive to curate at scale. These results suggest that data remains key to enabling better MCP use.

**Robustness lags far behind.** Table 2 demonstrates that pass@4 provides substantial gains, with `gpt-5-medium` and `claude-sonnet-4` achieving 68.50% and 44.88% compared to just 52.56% and 28.15% for pass@1. However, the performance at pass^4 drops sharply to 33.86% and 12.60%, respectively, underscoring the model's inconsistency and instability across runs. Similar discrepancies are observed across other models, with pass@4 often exceeding 30% while pass^4 remains in the 5% to 15% range, suggesting that while repeated attempts improve success, robustness under multi-turn tool use in MCP contexts remains a common challenge—a shortcoming that poses significant risks for real-world deployment where reliability across runs is essential.

Table 2: **Model comparison across MCPs.** Pass@1 is computed as the average over four independent runs, with the superscript showing the standard deviation; each MCP service value is also averaged over four runs. Within each model group (Proprietary / Open-Source), the best result is marked in **bold** and the second best result is underlined. For GPT-5 series models, explicit suffixes (e.g., "-medium") indicate the reasoning effort setting; for all models, results correspond to their default reasoning effort if supported. Abbreviations of MCP services are: FS = Filesystem, GH = GitHub, NT = Notion, PW = Playwright, PG = PostgreSQL.

| Model | MCP Services | | | | | Metrics | | |
|---|---|---|---|---|---|---|---|---|
| | FS | GH | NT | PW | PG | pass@1 | pass@4 | pass^4 |
| **Proprietary Models** | | | | | | | | |
| gpt-5-medium | **57.50** | **47.83** | **41.96** | **43.00** | **76.19** | **$52.56^{\pm1.29}$** | **68.50** | **33.86** |
| grok-4 | 50.83 | 14.13 | 2.68 | 35.00 | 58.33 | $31.69^{\pm2.91}$ | 44.88 | 18.11 |
| claude-opus-4.1 | 33.33 | 21.74 | 35.71 | 24.00 | 33.33 | $29.92^{\pm0.00}$ | – | – |
| claude-sonnet-4 | 27.50 | 16.30 | 21.43 | 26.00 | 53.57 | $28.15^{\pm2.57}$ | 44.88 | 12.60 |
| gpt-5-mini-medium | 33.33 | 18.48 | 16.07 | 12.00 | 61.90 | $27.36^{\pm3.12}$ | 45.67 | 9.45 |
| o3 | 35.83 | 14.13 | 24.11 | 15.00 | 36.90 | $25.39^{\pm2.04}$ | 43.31 | 12.60 |
| grok-code-fast-1 | 23.33 | 8.70 | 2.68 | 25.00 | 47.62 | $20.47^{\pm3.39}$ | 30.71 | 9.45 |
| qwen3-max | 10.83 | 14.13 | 16.96 | 8.00 | 44.05 | $17.72^{\pm1.31}$ | 22.83 | 11.02 |
| o4-mini | 25.00 | 14.13 | 20.54 | 12.00 | 11.90 | $17.32^{\pm2.30}$ | 31.50 | 6.30 |
| gemini-2.5-pro | 24.17 | 9.78 | 4.46 | 15.00 | 26.19 | $15.75^{\pm0.56}$ | 29.92 | 4.72 |
| gemini-2.5-flash | 8.33 | 15.22 | 6.25 | 6.00 | 10.71 | $9.06^{\pm0.68}$ | 18.11 | 3.94 |
| gpt-4.1 | 12.50 | 7.61 | 6.25 | 8.00 | 4.76 | $8.07^{\pm0.65}$ | 12.60 | 3.15 |
| gpt-5-nano-medium | 6.67 | 7.61 | 3.57 | 0.00 | 15.48 | $6.30^{\pm2.01}$ | 11.81 | 1.57 |
| gpt-4.1-mini | 3.33 | 6.52 | 1.79 | 0.00 | 9.52 | $3.94^{\pm0.96}$ | 7.09 | 1.57 |
| gpt-4.1-nano | 0.00 | 0.00 | 0.00 | 0.00 | 0.00 | $0.00^{\pm0.00}$ | 0.00 | 0.00 |
| **Open-Source Models** | | | | | | | | |
| qwen3-coder-plus | 13.33 | 19.57 | 19.64 | **30.00** | 47.62 | **$24.80^{\pm2.05}$** | **40.94** | **12.60** |
| kimi-k2-instruct | 14.17 | 16.30 | 8.04 | **30.00** | 47.62 | $21.85^{\pm1.16}$ | 31.50 | **12.60** |
| deepseek-v3.1 | **15.83** | 9.78 | 12.50 | 7.00 | 42.86 | $16.73^{\pm1.41}$ | 28.35 | 7.87 |
| glm-4.5 | 7.50 | **22.83** | **21.43** | 13.00 | 14.29 | $15.55^{\pm1.16}$ | 24.41 | 6.30 |
| gpt-oss-120b | 5.83 | 4.35 | 3.57 | 3.00 | 7.14 | $4.72^{\pm0.96}$ | 13.39 | 0.00 |

**More turns do not necessarily yield better performance.** Figure 4 highlights distinct tool-calling behaviors across models. In particular, the efficiency-accuracy correlation shows that stronger models succeed through better decision making and targeted exploration, not blind trial-and-error. Notably, kimi-k2-instruct often enters an *overcalling* mode, exceeding 30 turns with diminishing success rates—indicating the model might get stuck or loop without effective information retrieval. In contrast, gpt-5-medium achieves the highest pass@1 while maintaining reasonable turn budgets, demonstrating that success arises from efficient decision-making rather than exhaustive tool calls. Turn counts also vary significantly across MCP services (see Appendix G for details).

**Cost is not a proxy for performance.** Figure 21 shows that higher cost does not lead to higher accuracy. Some of the most expensive runs achieve lower pass@1, while several lower-cost runs reach stronger results. Table 9 reports per-task averages and further shows that costs vary widely even when the number of turns is similar. Higher cost alone does not imply better results.

## 3.3 BENCHMARKING MCP SERVERS AND AGENT FRAMEWORKS

Beyond measuring model performance, MCPMark can also be used to evaluate differences across MCP server implementations and agent frameworks. These aspects are important in practical MCP use, because the same model can behave very differently depending on server design or scaffold logic.

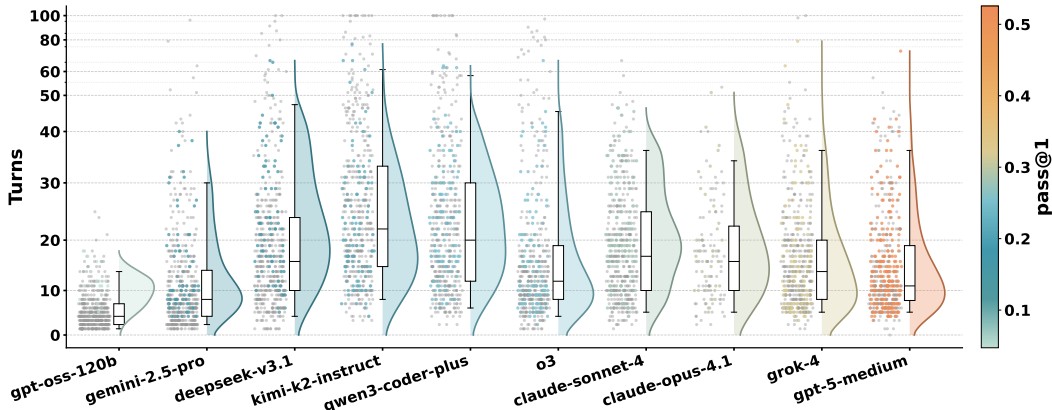

Figure 4: **Turns distribution.** Each point is one run (gray = fail). Plots show the turn distribution of successes; color encodes pass@1. Stronger models finish with fewer, better-targeted calls.

**MCP server variation.** As shown in Table 11, the same model can show large performance gaps across MCP servers. For GitHub tasks with `claude-sonnet-4`, the KlavisAI server reaches 31.5% pass@1, compared to 16.3% for the official server. A similar gap appears in Notion (34.8% vs. 21.4%). PostgreSQL shows the same trend on `claude-sonnet-4.5`: InsForge (54.8%) and Supabase (52.4%) both exceed the official server (48.8%). These gaps also align with differences in token usage, where higher-performing servers frequently require fewer tokens. Together, the results indicate that choices in schema exposure, error messaging, and other engineering details on the server side can materially influence agent success.

**Agent scaffold comparison.** We also benchmark `MCPMark-Agent` against `ReAct` (Yao et al., 2022) and `Codex` (OpenAI, 2025e) (Table 12). Surprisingly, naive iterative tool-calling provides the strongest baseline (`gpt-5-medium`: 52.6% pass@1), outperforming both `ReAct` (37.8%) and `Codex` (36.2%), suggesting that structured scaffolds may introduce redundant constraints in MCP settings.

## 4 ANALYSIS

In this section, we investigate two aspects that shape model performance on `MCPMark`: the role of reasoning effort in agent generalization, and the types of failures that prevent successful execution.

### 4.1 REASONING MODE AND EFFORT

We study how models benefit from different levels of reasoning effort, which are typically reflected in the number of consumed thinking tokens before issuing tool calls. Table 3 reports results for the `gpt-5` series and `claude-sonnet-4` across different effort settings.

**Model perspective.** The `gpt-5` series benefits from increased reasoning effort at moderate and large scales, though effects diverge by size. For `gpt-5`, medium effort raises pass@1 to 52.56% from 46.85% at low effort. `gpt-5-mini` shows even stronger relative gains, improving from 8.27% to 30.32% between low and high. By contrast, `gpt-5-nano` shows only marginal changes around 4% to 6%, suggesting models of this scale lack the capacity to exploit additional reasoning tokens. `claude-sonnet-4` is similarly insensitive, remaining stable around 27% to 28%. These results indicate that translating additional reasoning steps into better MCP use is non-trivial and likely depends on a model's base capacity and training approach.

**MCP perspective.** Reasoning effort selectively improves generalization in agentic tasks. Remote services benefit most: GitHub performance nearly doubles from 27.17% to 50.00% between low and high effort for `gpt-5`, while Notion rises from 36.61% to 44.64%. Local services remain stable, with PostgreSQL at 72% to 76% and Filesystem varying under 5 percentage points. We interpret this discrepancy as stemming from differences in training coverage. Remote services typically have limited exposure due to rate limits and access restrictions, making the tasks harder and

---

We also include results on other agent scaffolds in Table 13 for reference.

Table 3: **Reasoning effort.** Comparison of `gpt-5` series models and `claude-sonnet-4` under different reasoning effort settings. **Pass@1** is reported as mean with standard deviation (4 runs). Each model expands into its supported reasoning effort settings. Best values in each column are **bolded**.

| Model | Reasoning | Overall | 📁 FS | ⓞ GH | Ⓝ NT | 🐢 PW | 🐘 PG |
|---|---|---|---|---|---|---|---|
| gpt-5 | Low | $46.85^{\pm3.31}$ | $54.17^{\pm7.88}$ | $27.17^{\pm2.17}$ | $36.61^{\pm8.93}$ | $\mathbf{45.00}^{\pm2.00}$ | $73.81^{\pm4.76}$ |
| | Medium | $\mathbf{52.56}^{\pm1.29}$ | $\mathbf{57.50}^{\pm4.19}$ | $47.83^{\pm9.39}$ | $41.96^{\pm3.42}$ | $43.00^{\pm6.00}$ | $\mathbf{76.19}^{\pm8.69}$ |
| | High | $51.57^{\pm2.91}$ | $52.50^{\pm4.19}$ | $\mathbf{50.00}^{\pm2.51}$ | $\mathbf{44.64}^{\pm2.06}$ | $42.00^{\pm5.16}$ | $72.62^{\pm4.56}$ |
| gpt-5-mini | Low | $8.27^{\pm1.51}$ | $12.50^{\pm5.69}$ | $8.70^{\pm3.55}$ | $5.36^{\pm6.19}$ | $1.00^{\pm2.00}$ | $14.29^{\pm3.89}$ |
| | Medium | $27.36^{\pm3.60}$ | $33.33^{\pm7.20}$ | $18.48^{\pm8.96}$ | $16.07^{\pm6.84}$ | $12.00^{\pm7.30}$ | $61.90^{\pm6.73}$ |
| | High | $\mathbf{30.32}^{\pm1.98}$ | $\mathbf{35.00}^{\pm8.82}$ | $\mathbf{19.57}^{\pm2.51}$ | $\mathbf{20.54}^{\pm15.0}$ | $\mathbf{15.00}^{\pm6.00}$ | $\mathbf{66.67}^{\pm3.89}$ |
| gpt-5-nano | Low | $4.33^{\pm1.36}$ | $\mathbf{12.50}^{\pm4.19}$ | $0.00^{\pm0.00}$ | $0.00^{\pm0.00}$ | $0.00^{\pm0.00}$ | $8.33^{\pm4.56}$ |
| | Medium | $\mathbf{6.30}^{\pm2.32}$ | $6.67^{\pm6.09}$ | $7.61^{\pm2.17}$ | $\mathbf{3.57}^{\pm0.00}$ | $0.00^{\pm0.00}$ | $\mathbf{15.48}^{\pm5.99}$ |
| | High | $5.12^{\pm2.36}$ | $5.83^{\pm5.69}$ | $\mathbf{8.70}^{\pm3.55}$ | $0.89^{\pm1.79}$ | $\mathbf{2.00}^{\pm2.31}$ | $9.52^{\pm3.89}$ |
| claude-sonnet-4 | N/A | $28.15^{\pm2.97}$ | $\mathbf{27.50}^{\pm3.19}$ | $16.30^{\pm6.52}$ | $21.43^{\pm5.83}$ | $\mathbf{26.00}^{\pm6.93}$ | $\mathbf{53.57}^{\pm7.14}$ |
| | Low | $27.36^{\pm1.97}$ | $23.33^{\pm5.44}$ | $25.00^{\pm4.16}$ | $\mathbf{22.32}^{\pm3.42}$ | $22.00^{\pm4.00}$ | $48.81^{\pm8.13}$ |
| | High | $\mathbf{28.35}^{\pm2.73}$ | $23.33^{\pm4.71}$ | $\mathbf{28.26}^{\pm2.51}$ | $19.64^{\pm9.45}$ | $\mathbf{26.00}^{\pm2.31}$ | $50.00^{\pm8.25}$ |

Figure 5: **Failure breakdown across models.** Failures are categorized as either *implicit* (task completes normally but fails verification) or explicit (e.g., *context window overflow, turn limit exceeded, abandoned, premature stop,* or *malformed tool calls*).

requiring stronger generalization at test-time. Reasoning helps bridge this gap by enabling models to extrapolate to unseen cases, aligning with recent discussions (Yao et al., 2023b; Yao, 2025) that highlight ***"language generalizes through reasoning in agents"***.

## 4.2 FAILURE BREAKDOWN

**Introduction.** We classify failures into two categories to ease presentation: *implicit* and *explicit*. Implicit failures occur when the task completes successfully but the output does not meet the required specifications. These often stem from issues such as reasoning errors, suboptimal planning, ineffective tool usage, or difficulty handling long contexts, which may interact to cause complex failures that are difficult to attribute to a single factor. In contrast, explicit failures can be directly linked to specific issues. These include *context window overflow* (input exceeding the model's processing length), *turn limit exceeded* (the model exhausts its allowed interaction steps), *abandoned* tasks (model decides the task is infeasible), *premature stop* (model halts without completing or making necessary tool calls), and *malformed tool calls* (invalid parameters or improperly structured payloads).

**Observations.** As seen in Figure 5, implicit failures account for the majority of errors across all models, often exceeding 50%. Models like `gpt-5-high` and `kimi-k2-instruct` show over 80% implicit failures, indicating they generally complete tasks without obvious breakdowns, with errors being more subtle and capability-driven. In contrast, `gemini-2.5-flash` and `gpt-4.1` have lower implicit failure rates (52% and 66%, respectively), suggesting more explicit causes. For explicit failures, `gemini-2.5-flash` and `gpt-4.1` mainly experience *abandoned* or *premature stop* errors, reflecting weaker reasoning and planning. `gemini-2.5-flash` also shows a higher incidence of *malformed tool calls* (around 10%), possibly due to mismatches in tool-call conventions or

insufficient training. `gpt-5-high` has more *context window overflow* errors, indicating difficulties with long-context handling, while `kimi-k2-instruct` faces frequent *turn limit exceeded* errors, often due to repetitive tool-calling loops. These results suggest that explicit errors are model-specific, highlighting the need for targeted improvements in reasoning, context management, and tool use.

## 5 RELATED WORK

**LLM Agents**. With the development of large language models (LLMs) (Team, 2025; Anthropic, 2025a; Team et al., 2025; OpenAI, 2025c; Comanici et al., 2025), LLM agents have progressed from early prompting methods such as ReAct (Yao et al., 2023b), which integrated reasoning traces with tool actions, to more structured designs like MetaGPT (Hong et al., 2024) that coordinate multi-agent collaboration through explicit role assignment. This evolution has been supported by research on tool use (Schick et al., 2023; Qin et al., 2023; Patil et al., 2024), which explore when and how models should call APIs, as well as planning and reflection methods (Yao et al., 2023a; Shinn et al., 2023; Wang et al., 2024a) that improve robustness in multi-step workflows. Multi-agent frameworks (Wu et al., 2024; Li et al., 2023; Chen et al., 2023) further demonstrate the benefits of coordinated division of labor. In applied domains, coding agents (Yang et al., 2024; Wang et al., 2024b) enable real repository interaction; GUI and computer-use agents are advanced by benchmarks (Zhou et al., 2023; Deng et al., 2023; Xie et al., 2024); and deep research efforts are represented by initiatives (Wei et al., 2025; Starace et al., 2025; Du et al., 2025). Together, these developments illustrate the trend toward general agents that can operate across heterogeneous systems and contexts, naturally pointing to the need for standardized protocols such as the Model Context Protocol (MCP) (Anthropic, 2024) that provide a unifying interface for tool and environment integration.

**Benchmarks for evaluating MCP use**. Recent work has begun to systematically benchmark agent performance in MCP-enabled settings (Yan et al., 2025; Liu et al., 2025; Mo et al., 2025; Gao et al., 2025). MCP-Universe (Luo et al., 2025) constructed tasks across multiple domains and evaluators, revealing the difficulty models face with long and dynamic workflows. LiveMCP-101 (Yin et al., 2025) focused on multi-tool interaction and execution-plan validation, while MCP-AgentBench (Guo et al., 2025) scaled up evaluation with hundreds of tasks spanning diverse servers and tools. These efforts primarily emphasize broad tool coverage or easier execution but leave gaps in assessing high-fidelity workflows tied to realistic application environments. Our proposed `MCPMark` addresses this by designing tasks with diverse CRUD operations in containerized settings to ensure safety and reproducibility. Each task is paired with programmatic verification scripts and full environment state tracking, enabling reliable and fine-grained evaluation.

## 6 DISCUSSION ON LIMITATIONS AND FUTURE DIRECTIONS

Our task creation pipeline, while ensuring task quality, is difficult to scale. This creates a bottleneck for producing the large-scale training data needed to advance the field. Furthermore, the steep difficulty of many tasks in `MCPMark` limits its utility for evaluating and guiding the development of smaller, more efficient models. Future work on the benchmark should therefore focus on introducing a more fine-grained difficulty gradient, potentially through semi-automated task generation and a reduced task execution chain. Additionally, to better reflect real-world complexity, the benchmark could be expanded to include tasks with ambiguous user intent. This would test an agent's ability to ask clarifying questions or infer the user's actual intent. Finally, incorporating a wider variety of MCP servers could also help challenge agents across a more diverse set of digital tools.

## ETHICS STATEMENT

This section outlines how we address the ethical considerations involved in the construction of our benchmark, which includes several key components that could raise ethical concerns:

- **Initial State of MCP Environment:** Each initial state and environment used in the benchmark is provided with the appropriate license information (see Appendix H for details). A few environments were self-curated, and for these, we have ensured transparency and compliance with relevant licensing requirements, promoting ethical usage.
- **Task Curation:** All tasks included in the benchmark were collaboratively annotated by both experts and AI agents. The experts involved in the curation process have been properly recognized as co-authors in the author list, ensuring that their contributions are duly acknowledged. Additionally, the licenses for the agents used, including Claude Code (License) and Cursor (License), are provided to ensure that all resources are used responsibly and in accordance with the relevant licensing terms for research purposes.
- **MCP Servers:** The licenses for each specific MCP server used in the benchmark are provided in Appendix C. This ensures that all external systems and tools are properly licensed for research and evaluation purposes.

By adhering to these practices, we ensure that high ethical standards are maintained throughout the construction of the benchmark, and that all resources are used responsibly and in accordance with relevant regulations.

## REPRODUCIBILITY STATEMENT

In order to ensure the reproducibility of our experiments, we have made available the evaluation code, task data, and corresponding run instructions in the supplementary materials. The evaluation code has been modified to remove any identifiable personal information to protect privacy. Additionally, the tasks and data used for evaluation are included in the supplementary materials along with detailed instructions on how to execute the experiments. This ensures that other researchers can replicate our results under the same conditions while adhering to privacy standards.

## ACKNOWLEDGEMENTS

This project was partially supported by the Singapore Ministry of Education Academic Research Fund Tier 1 (Award Number: T1 251RES2514).

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

# APPENDIX

## TABLE OF CONTENTS

## A   USE OF LLMS

Large language models were used as general-purpose assistants to support writing, debugging, and the generation of some initial environment states. Specifically, claude-code and codex were prompted to produce structured directory layouts, starter scripts, or database schemas that served as initial configurations in selected tasks. These outputs were carefully reviewed and refined by human experts to ensure correctness, realism, and alignment with benchmark goals.

LLMs also helped improve the grammar, flow of writing, and occasionally assisted in resolving minor coding issues such as syntax errors or implementation quirks. All core ideas, benchmark designs, implementations, experiments, and analyses were independently developed by the authors. No part of the scientific contribution or methodological reasoning was generated by an LLM.

## B  DETAILS OF THE TASK CREATION PIPELINE

We use Playwright as an example to illustrate the guideline for human experts and the initial instruction/prompt for the task creation agent. These are simplified for reference.

---

**Guideline (Playwright)**

Step 1. **Select the starting environment**
Pick a website or web app as the initial state. Prefer a staging or test instance to avoid side effects. Examples: a Reddit-like forum or a Shopping Admin dashboard.

Step 2. **Configure the agent environment**
In Cursor or Claude Code, set up the MCP server stack and include the *Playwright MCP server* so the agent can control a browser.

Step 3. **Define an initial question or topic**
Write a `seed question or topic` that will guide agent exploration and task creation. It can be broad or moderately specific.

Step 4. **Create and refine the task**

 Step 4.1. **Exploration with the agent**
 Have the agent read the initial instruction (which includes the seed question), then explore the target site together with the agent. Based on the collected context, ask the agent to propose a task that fits the objectives and requirements.

 Step 4.2. **Provide feedback to improve the task**
 Guide the agent to revise the task as needed. Examples:
  • If verification is weak: *"This task is not sufficiently verifiable. Please revise it to make verification clearer and more reliable."*
  • If exploration lacks coverage: *"You can explore deeper to collect more diverse and detailed information."*
  • If subtasks feel disconnected: *"Make the subtasks integrated rather than unrelated."*

 Step 4.3. **Save the task**
 Store the *task description* and the *verification script* as separate files. Use a consistent folder structure based on category and name. Follow well-structured prior examples for formatting.

 Step 4.4. **Human-in-the-loop adjustments**
 Iterate between the agent and the reviewer until both the task description and the verification script meet quality standards.

Step 5. **Execute and verify**
Run the task with Playwright MCP to reach the final state, then run the verification script. Stress-test the checker to confirm:

 Step 5.1. The task is executable end to end.

 Step 5.2. Pass or fail is clear and objective.

 Step 5.3. The script flags both correct and incorrect outcomes, including edge cases.

Step 6. **Assess difficulty (optional)**
If the task and checker pass, consider whether difficulty is high enough to test the model. Adjust scope or constraints if needed.

**Notes.** These steps target experts working with Cursor or Claude Code. They are guidelines. If issues appear, collaborate with colleagues to debug efficiently.

---

**Initial Instruction for Task Creation Agent (Playwright)**

**Your job is to:**

1. First explore the web environment to understand available MCP tools and capabilities.

2. Generate **one** challenging, verifiable, and realistic task based on your collected information.

3. Focus your exploration and task generation on the following specific topic or question:

   - Use this as a guiding theme for creating more targeted and relevant tasks.
   - Ensure the task addresses different aspects or components related to this requirement.

**Playwright MCP Tools Reference:**

```
<playwright_mcp_doc>
[contents of docs/playwright-mcp-introduction.md go here]
</playwright_mcp_doc>
```

**Output Format:**

```
{
  "tasks": [
    {
      "task_id": "task_1",
      "description": "Clear, conversational task description",
      "difficulty": "hard",
      "verification_criteria": ["criterion 1", "criterion 2"],
      "expected_mcp_calls": ["browser_navigate", "browser_snapshot",
          "browser_click"],
      "estimated_complexity": "high"
    }
  ]
}
```

Based on the given web application environment, write **one** challenging, verifiable, and realistic browser automation task that aligns with users' actual web interaction workflows. The goal is to evaluate an Agent's ability to use Playwright MCP tools effectively. **Requirements:**

- **Difficulty**: The task should be really hard ... (omitted)

- **Verifiability**: Avoid open-ended outcomes ... (omitted)

- **Authenticity**: Describe the task in a natural, conversational tone ... (omitted)

- **Context Awareness**: Leverage page structure, form elements, navigation patterns, ... (omitted)

**Start by exploring the web application environment using MCP tools** to understand the current structure, interactive elements, and user workflows, then generate a task that combines:

1. Your real-time MCP exploration findings.

2. The specific website structure and interactive elements you discover.

3. A focus on browser automation operations that require multiple Playwright MCP tools rather than only content reading.

4. The specific focus area: `<seed_topic>`.

**Please explore thoroughly before creating the task. Consider:**

- Form elements and input fields.

- Navigation patterns and menu structures.

- Dynamic content and interactive features.

- User workflow patterns.

- Authentication and session management.

- Data submission and validation processes.

## C  MCP SERVERS

We relied on five Model Context Protocol (MCP) servers in our setup. Below we summarize their functionality, invocation, repository, and license.

**Filesystem.** The filesystem server provides local read, write, and directory operations over the host file system. It is invoked as `@modelcontextprotocol/server-filesystem`. The implementation is hosted at github.com/modelcontextprotocol/servers under the MIT License.

**GitHub.** The GitHub server integrates with the GitHub API to manage repositories, issues, and pull requests. The endpoint used is `https://api.githubcopilot.com/mcp/`. The code is available at github.com/github/github-mcp-server, released under the MIT License.

**Notion.** The Notion server allows interaction with Notion databases and pages. It is invoked as `@notionhq/notion-mcp-server`. The official repository is github.com/makenotion/notion-mcp-server, licensed under the MIT License.

**Playwright.** The Playwright server enables browser automation and scripted web workflows. It is started using `@playwright/mcp@latest`. The source code is provided at github.com/microsoft/playwright-mcp, distributed under the Apache License 2.0.

**PostgreSQL.** The PostgreSQL server provides access to a relational database through SQL queries. It is launched with `postgres-mcp -access-mode=unrestricted`. The implementation is maintained at github.com/crystaldba/postgres-mcp, and is released under the MIT License.

# D    DETAILED MCP BENCHMARK RESULTS

Tables 2 and 9 presented the overall success rates and usage statistics, aggregated across all MCPs. Here we provide the corresponding breakdown by individual MCP from Table 4 to Table 8. #Input and #Output are measured in thousands of tokens (K), and Cost is reported in USD. For success metrics, **bold** and underline indicate the best and second-best results, respectively. For usage statistics, **bold** and underline denote the largest and second-largest values, without implying better performance.

Table 4: 📁 Filesystem  MCP benchmark results.

| Model | Metrics | | | Per-Task Avg Usage | | | | |
|---|---|---|---|---|---|---|---|---|
| | Pass@1 | Pass@4 | Pass^4 | # Input | # Output | Cost | Turns | Tool Calls |
| 🔒 Proprietary Models | | | | | | | | |
| 🌀 gpt-5-medium | **57.50**$^{\pm 3.63}$ | **76.67** | **36.67** | 215.96 | 17.38 | 0.44 | 10.06 | 21.07 |
| ⊘ grok-4 | 50.83$^{\pm 6.40}$ | 73.33 | 26.67 | 247.33 | 10.70 | 0.90 | 10.80 | 16.87 |
| 🌀 o3 | 35.83$^{\pm 2.76}$ | 50.00 | 26.67 | **689.64** | 17.79 | 1.52 | **28.79** | 27.80 |
| 🌀 gpt-5-mini-medium | 33.33$^{\pm 6.24}$ | 53.33 | 10.00 | 398.34 | 12.58 | 0.12 | 14.84 | **36.93** |
| ✳ claude-opus-4.1 | 33.33$^{\pm 0.00}$ | – | – | 272.17 | 4.37 | **4.41** | 16.37 | 15.40 |
| ✳ claude-sonnet-4 | 27.50$^{\pm 2.76}$ | 50.00 | 6.67 | 302.21 | 4.00 | 0.97 | 16.02 | 15.08 |
| 🌀 o4-mini | 25.00$^{\pm 2.89}$ | 36.67 | 13.33 | 293.34 | 15.89 | 0.39 | 20.88 | 19.88 |
| ✦ gemini-2.5-pro | 24.17$^{\pm 3.63}$ | 43.33 | 10.00 | 214.97 | 7.75 | 0.65 | 14.35 | 14.72 |
| ⊘ grok-code-fast-1 | 23.33$^{\pm 7.45}$ | 40.00 | 10.00 | 276.40 | 2.36 | 0.06 | 16.38 | 16.77 |
| 🌀 gpt-4.1 | 12.50$^{\pm 1.44}$ | 20.00 | 3.33 | 143.95 | 1.81 | 0.30 | 9.28 | 18.48 |
| ✦ gemini-2.5-flash | 8.33$^{\pm 1.67}$ | 13.33 | 6.67 | 67.64 | 7.57 | 0.04 | 6.50 | 11.15 |
| 🌀 gpt-5-nano-medium | 6.67$^{\pm 5.27}$ | 16.67 | 0.00 | 462.74 | **19.53** | 0.03 | 20.75 | 27.76 |
| 🌀 gpt-4.1-mini | 3.33$^{\pm 0.00}$ | 3.33 | 3.33 | 196.15 | 1.63 | 0.08 | 15.50 | 19.57 |
| 🌀 gpt-4.1-nano | 0.00$^{\pm 0.00}$ | 0.00 | 0.00 | 116.98 | 1.32 | 0.01 | 12.17 | 15.32 |
| 🤗 Open-Source Models | | | | | | | | |
| 🐋 deepseek-v3.1 | **15.83**$^{\pm 1.44}$ | **26.67** | 6.67 | 421.33 | 3.38 | 0.24 | 23.83 | 23.12 |
| 🌐 kimi-k2-instruct-0905 | 14.17$^{\pm 1.44}$ | 23.33 | 6.67 | 696.79 | **4.47** | 0.43 | 26.27 | 25.70 |
| 🐦 qwen3-coder-plus | 13.33$^{\pm 6.67}$ | **26.67** | 3.33 | **972.41** | 4.15 | 0.20 | **28.23** | **27.32** |
| 🐦 qwen3-max | 10.83$^{\pm 1.44}$ | 13.33 | **10.00** | 389.56 | 2.87 | **0.48** | 19.27 | 18.39 |
| ❷ glm-4.5 | 7.50$^{\pm 1.44}$ | 13.33 | 3.33 | 193.95 | 3.92 | 0.07 | 16.39 | 17.09 |
| 🌀 gpt-oss-120b | 5.83$^{\pm 4.33}$ | 16.67 | 0.00 | 19.75 | 1.08 | < 0.01 | 4.62 | 3.62 |

Table 5: ⬤ GitHub MCP benchmark results.

| Model | Metrics | | | Per-Task Avg Usage | | | | |
|---|---|---|---|---|---|---|---|---|
| | Pass@1 | Pass@4 | Pass^4 | # Input | # Output | Cost | Turns | Tool Calls |
| 🔒 Proprietary Models | | | | | | | | |
| ⬤ gpt-5-medium | $\mathbf{47.83}^{\pm 8.13}$ | **65.22** | **17.39** | 659.73 | 20.57 | 1.03 | 14.33 | **21.23** |
| ✳ claude-opus-4.1 | $21.74^{\pm 0.00}$ | – | – | 620.63 | 5.84 | **9.75** | 10.78 | 10.13 |
| ⬤ gpt-5-mini-medium | $18.48^{\pm 7.76}$ | 34.78 | 4.35 | 614.68 | 7.71 | 0.17 | 13.92 | 17.28 |
| ✳ claude-sonnet-4 | $16.30^{\pm 5.65}$ | 30.43 | 8.70 | 696.81 | 4.44 | 2.16 | 11.16 | 10.50 |
| ✦ gemini-2.5-flash | $15.22^{\pm 2.17}$ | 21.74 | 8.70 | **1107.04** | 12.70 | 0.36 | 10.46 | 17.71 |
| ⬤ grok-4 | $14.13^{\pm 3.61}$ | 21.74 | 8.70 | 804.50 | 1.93 | 2.44 | 12.98 | 16.76 |
| ⬤ o4-mini | $14.13^{\pm 6.43}$ | 26.09 | 4.35 | 510.13 | 8.74 | 0.60 | 10.92 | 10.08 |
| ⬤ o3 | $14.13^{\pm 3.61}$ | 21.74 | 4.35 | 451.18 | 3.56 | 0.93 | 9.20 | 8.24 |
| ✦ gemini-2.5-pro | $9.78^{\pm 1.88}$ | 21.74 | 0.00 | 173.43 | 5.75 | 0.52 | 5.45 | 6.29 |
| ⬤ grok-code-fast-1 | $8.70^{\pm 5.32}$ | 17.39 | 4.35 | 751.41 | 6.50 | 0.16 | **17.85** | 17.28 |
| ⬤ gpt-5-nano-medium | $7.61^{\pm 1.88}$ | 13.04 | 0.00 | 751.62 | **26.77** | 0.05 | 15.15 | 17.63 |
| ⬤ gpt-4.1 | $7.61^{\pm 1.88}$ | 8.70 | 4.35 | 445.88 | 2.49 | 0.91 | 9.95 | 14.97 |
| ⬤ gpt-4.1-mini | $6.52^{\pm 6.52}$ | 17.39 | 0.00 | 466.70 | 1.51 | 0.19 | 12.00 | 14.63 |
| ⬤ gpt-4.1-nano | $0.00^{\pm 0.00}$ | 0.00 | 0.00 | 312.86 | 2.59 | 0.03 | 9.27 | 11.04 |
| 👶 Open-Source Models | | | | | | | | |
| ⬤ glm-4.5 | $\mathbf{22.83}^{\pm 6.43}$ | **34.78** | **13.04** | 482.00 | 3.65 | 0.16 | 11.92 | 11.04 |
| ⬤ qwen3-coder-plus | $19.57^{\pm 6.52}$ | **34.78** | **13.04** | **1987.14** | 3.36 | 0.40 | 19.12 | 18.13 |
| ⬤ kimi-k2-instruct-0905 | $16.30^{\pm 1.88}$ | 26.09 | 8.70 | 995.65 | **8.25** | 0.62 | 23.68 | 23.23 |
| ⬤ qwen3-max | $14.13^{\pm 3.61}$ | 17.39 | 4.35 | 1348.13 | 2.55 | **1.63** | **26.70** | **25.78** |
| ⬤ deepseek-v3.1 | $9.78^{\pm 1.88}$ | 13.04 | 8.70 | 362.36 | 2.24 | 0.21 | 9.46 | 9.22 |
| ⬤ gpt-oss-120b | $4.35^{\pm 3.07}$ | 8.70 | 0.00 | 76.30 | 1.41 | < 0.01 | 4.62 | 3.62 |

Table 6: N Notion  MCP benchmark results.

| Model | Metrics | | | Per-Task Avg Usage | | | | |
|---|---|---|---|---|---|---|---|---|
| | Pass@1 | Pass@4 | Pass^4 | # Input | # Output | Cost | Turns | Tool Calls |
| 🔒 Proprietary Models | | | | | | | | |
| 🌀 gpt-5-medium | **41.96**$^{\pm2.96}$ | **50.00** | **32.14** | 375.04 | 31.62 | 0.79 | 12.94 | 21.60 |
| ✳ claude-opus-4.1 | 35.71$^{\pm0.00}$ | – | – | 638.06 | 3.93 | **9.87** | 17.04 | 16.04 |
| 🟡 o3 | 24.11$^{\pm3.89}$ | 46.43 | 7.14 | 224.93 | 9.47 | 0.53 | 13.72 | 12.72 |
| ✳ claude-sonnet-4 | 21.43$^{\pm5.05}$ | 39.29 | 7.14 | 646.64 | 4.24 | 2.00 | 19.71 | 18.71 |
| 🟡 o4-mini | 20.54$^{\pm5.85}$ | 42.86 | 7.14 | 267.63 | 25.97 | 0.41 | 15.29 | 14.29 |
| 🌀 gpt-5-mini-medium | 16.07$^{\pm5.92}$ | 32.14 | 3.57 | **705.09** | 12.34 | 0.20 | 14.60 | 17.28 |
| ✦ gemini-2.5-flash | 6.25$^{\pm4.64}$ | 21.43 | 0.00 | 201.00 | 6.58 | 0.08 | 6.11 | 9.61 |
| 🌀 gpt-4.1 | 6.25$^{\pm1.55}$ | 14.29 | 0.00 | 135.55 | 1.37 | 0.28 | 8.58 | 11.82 |
| ✦ gemini-2.5-pro | 4.46$^{\pm2.96}$ | 7.14 | 0.00 | 212.92 | 7.13 | 0.64 | 7.12 | 8.67 |
| 🌀 gpt-5-nano-medium | 3.57$^{\pm0.00}$ | 3.57 | 3.57 | 204.32 | **32.08** | 0.02 | 7.46 | 8.74 |
| ⓔ grok-4 | 2.68$^{\pm1.55}$ | 3.57 | 0.00 | 678.64 | 13.04 | 2.23 | 20.14 | **24.80** |
| ⓔ grok-code-fast-1 | 2.68$^{\pm1.55}$ | 3.57 | 0.00 | 561.49 | 7.26 | 0.12 | **20.27** | 20.09 |
| 🌀 gpt-4.1-mini | 1.79$^{\pm1.79}$ | 3.57 | 0.00 | 262.75 | 1.35 | 0.11 | 12.57 | 14.56 |
| 🌀 gpt-4.1-nano | 0.00$^{\pm0.00}$ | 0.00 | 0.00 | 93.38 | 1.40 | < 0.01 | 9.64 | 10.93 |
| 🧑 Open-Source Models | | | | | | | | |
| Ⓩ glm-4.5 | **21.43**$^{\pm2.53}$ | 32.14 | **10.71** | 625.97 | 5.04 | 0.21 | 22.15 | 21.17 |
| ⓥ qwen3-coder-plus | 19.64$^{\pm6.44}$ | **39.29** | 7.14 | 796.73 | 2.75 | 0.16 | 21.07 | 20.23 |
| ⓥ qwen3-max | 16.96$^{\pm4.64}$ | 25.00 | 3.57 | 973.92 | 3.66 | **1.19** | 26.57 | 25.63 |
| ⓦ deepseek-v3.1 | 12.50$^{\pm3.09}$ | 28.57 | 0.00 | 503.35 | 2.20 | 0.29 | 17.94 | 17.40 |
| ⓢ kimi-k2-instruct-0905 | 8.04$^{\pm2.96}$ | 10.71 | 3.57 | **1117.21** | **5.20** | 0.68 | **33.55** | **32.72** |
| 🌀 gpt-oss-120b | 3.57$^{\pm2.53}$ | 14.29 | 0.00 | 68.31 | 1.72 | < 0.01 | 5.49 | 4.49 |

Table 7: 🎭 Playwright  MCP benchmark results.

| Model | Metrics | | | Per-Task Avg Usage | | | | |
|---|---|---|---|---|---|---|---|---|
| | Pass@1 | Pass@4 | Pass^4 | # Input | # Output | Cost | Turns | Tool Calls |
| 🔒 Proprietary Models | | | | | | | | |
| 🌀 gpt-5-medium | **43.00**$^{\pm 5.20}$ | **56.00** | **36.00** | 1807.17 | **21.79** | 2.48 | 23.78 | 22.96 |
| ⊘ grok-4 | 35.00$^{\pm 7.68}$ | 48.00 | 20.00 | 1264.91 | 6.64 | 3.89 | 20.05 | 23.02 |
| ✳ claude-sonnet-4 | 26.00$^{\pm 6.00}$ | 36.00 | 8.00 | 1241.92 | 3.52 | 3.78 | 19.80 | 19.12 |
| ⊘ grok-code-fast-1 | 25.00$^{\pm 1.73}$ | 36.00 | 8.00 | 1157.72 | 7.17 | 0.24 | 18.23 | 18.18 |
| ✳ claude-opus-4.1 | 24.00$^{\pm 0.00}$ | – | – | 1146.05 | 2.88 | **17.41** | 19.04 | 18.40 |
| ✦ gemini-2.5-pro | 15.00$^{\pm 1.73}$ | 32.00 | 4.00 | 1696.44 | 5.58 | 4.32 | 19.15 | 18.33 |
| 🌀 o3 | 15.00$^{\pm 5.20}$ | 32.00 | 8.00 | 556.30 | 4.46 | 1.15 | 16.30 | 15.40 |
| 🌀 o4-mini | 12.00$^{\pm 2.83}$ | 28.00 | 0.00 | 862.51 | 18.07 | 1.03 | 17.70 | 16.93 |
| 🌀 gpt-5-mini-medium | 12.00$^{\pm 6.32}$ | 24.00 | 4.00 | 1814.94 | 8.55 | 0.47 | 22.75 | 22.04 |
| 🌀 gpt-4.1 | 8.00$^{\pm 2.83}$ | 12.00 | 4.00 | 859.77 | 0.86 | 1.73 | 13.80 | 15.21 |
| ✦ gemini-2.5-flash | 6.00$^{\pm 2.00}$ | 12.00 | 0.00 | 3838.93 | 8.21 | 1.17 | 26.33 | **38.78** |
| 🌀 gpt-5-nano-medium | 0.00$^{\pm 0.00}$ | 0.00 | 0.00 | 711.95 | 17.71 | 0.04 | 18.52 | 17.55 |
| 🌀 gpt-4.1-mini | 0.00$^{\pm 0.00}$ | 0.00 | 0.00 | **4959.14** | 3.28 | 1.99 | **31.33** | 31.52 |
| 🌀 gpt-4.1-nano | 0.00$^{\pm 0.00}$ | 0.00 | 0.00 | 389.80 | 0.74 | 0.04 | 13.51 | 13.61 |
| 😀 Open-Source Models | | | | | | | | |
| 🐋 qwen3-coder-plus | **30.00**$^{\pm 4.47}$ | **48.00** | 8.00 | 2851.57 | 2.39 | 0.57 | 21.21 | 20.40 |
| 🌑 kimi-k2-instruct-0905 | **30.00**$^{\pm 6.00}$ | 40.00 | 20.00 | 1358.02 | 2.17 | 0.82 | 20.64 | 19.79 |
| ⓩ glm-4.5 | 13.00$^{\pm 3.32}$ | 20.00 | 4.00 | 582.73 | **2.76** | 0.20 | 15.36 | 14.61 |
| 🐋 qwen3-max | 8.00$^{\pm 0.00}$ | 12.00 | 4.00 | 2297.67 | 1.16 | **2.76** | **27.83** | **27.41** |
| 🐳 deepseek-v3.1 | 7.00$^{\pm 3.32}$ | 16.00 | 0.00 | 836.01 | 1.77 | 0.47 | 19.09 | 20.78 |
| 🌀 gpt-oss-120b | 3.00$^{\pm 1.73}$ | 4.00 | 0.00 | 139.33 | 1.27 | 0.01 | 7.21 | 6.26 |

Table 8: 🐘 PostgreSQL  MCP benchmark results.

| Model | Metrics | | | Per-Task Avg Usage | | | | |
|---|---|---|---|---|---|---|---|---|
| | Pass@1 | Pass@4 | Pass^4 | # Input | # Output | Cost | Turns | Tool Calls |
| 🔒 Proprietary Models | | | | | | | | |
| 🌀 gpt-5-medium | **76.19**$^{\pm7.53}$ | **100.00** | **47.62** | 113.35 | 17.04 | 0.31 | 13.37 | 12.45 |
| 🌀 gpt-5-mini-medium | 61.90$^{\pm5.83}$ | 90.48 | 28.57 | 115.40 | 9.27 | 0.05 | 11.77 | 10.77 |
| ⓧ grok-4 | 58.33$^{\pm7.81}$ | 80.95 | 38.10 | 186.07 | 8.23 | 0.68 | 17.89 | 17.08 |
| ✳ claude-sonnet-4 | 53.57$^{\pm6.19}$ | 71.43 | 38.10 | **331.10** | 7.54 | 1.11 | **26.80** | **25.81** |
| ⓧ grok-code-fast-1 | 47.62$^{\pm4.76}$ | 61.90 | 28.57 | 226.41 | 5.46 | 0.05 | 19.70 | 18.70 |
| 🟡 o3 | 36.90$^{\pm3.95}$ | 66.67 | 14.29 | 63.56 | 4.72 | 0.16 | 10.71 | 9.71 |
| ✳ claude-opus-4.1 | 33.33$^{\pm0.00}$ | – | – | 260.68 | 9.80 | **4.64** | 24.86 | 23.86 |
| ✦ gemini-2.5-pro | 26.19$^{\pm7.90}$ | 47.62 | 9.52 | 39.74 | 8.91 | 0.23 | 7.45 | 6.45 |
| 🌀 gpt-5-nano-medium | 15.48$^{\pm5.19}$ | 28.57 | 4.76 | 105.02 | **23.04** | 0.01 | 9.46 | 10.15 |
| 🟡 o4-mini | 11.90$^{\pm4.12}$ | 19.05 | 4.76 | 15.92 | 5.76 | 0.04 | 5.06 | 4.06 |
| ✦ gemini-2.5-flash | 10.71$^{\pm6.19}$ | 23.81 | 4.76 | 46.08 | 9.93 | 0.04 | 8.76 | 11.38 |
| 🌀 gpt-4.1-mini | 9.52$^{\pm3.37}$ | 14.29 | 4.76 | 46.63 | 1.78 | 0.02 | 9.77 | 11.61 |
| 🌀 gpt-4.1 | 4.76$^{\pm0.00}$ | 4.76 | 4.76 | 55.11 | 1.20 | 0.12 | 8.12 | 10.54 |
| 🌀 gpt-4.1-nano | 0.00$^{\pm0.00}$ | 0.00 | 0.00 | 71.06 | 2.43 | < 0.01 | 8.73 | 10.18 |
| 🤗 Open-Source Models | | | | | | | | |
| 🌀 qwen3-coder-plus | **47.62**$^{\pm5.83}$ | 61.90 | **38.10** | **573.90** | 5.13 | 0.12 | 29.00 | 28.00 |
| 🌑 kimi-k2-instruct-0905 | **47.62**$^{\pm4.76}$ | **66.67** | 28.57 | 441.16 | **5.38** | 0.28 | **30.21** | **29.25** |
| 🌀 qwen3-max | 44.05$^{\pm2.06}$ | 52.38 | **38.10** | 192.13 | 4.91 | 0.26 | 18.88 | 17.92 |
| 🐳 deepseek-v3.1 | 42.86$^{\pm7.53}$ | 61.90 | 28.57 | 316.60 | 4.65 | 0.19 | 26.48 | 25.49 |
| ⓩ glm-4.5 | 14.29$^{\pm7.53}$ | 23.81 | 0.00 | 204.61 | 5.14 | 0.07 | 25.39 | 24.40 |
| 🌀 gpt-oss-120b | 7.14$^{\pm2.38}$ | 23.81 | 0.00 | 21.36 | 1.42 | < 0.01 | 5.07 | 4.07 |

# E CASE STUDIES BY MCP

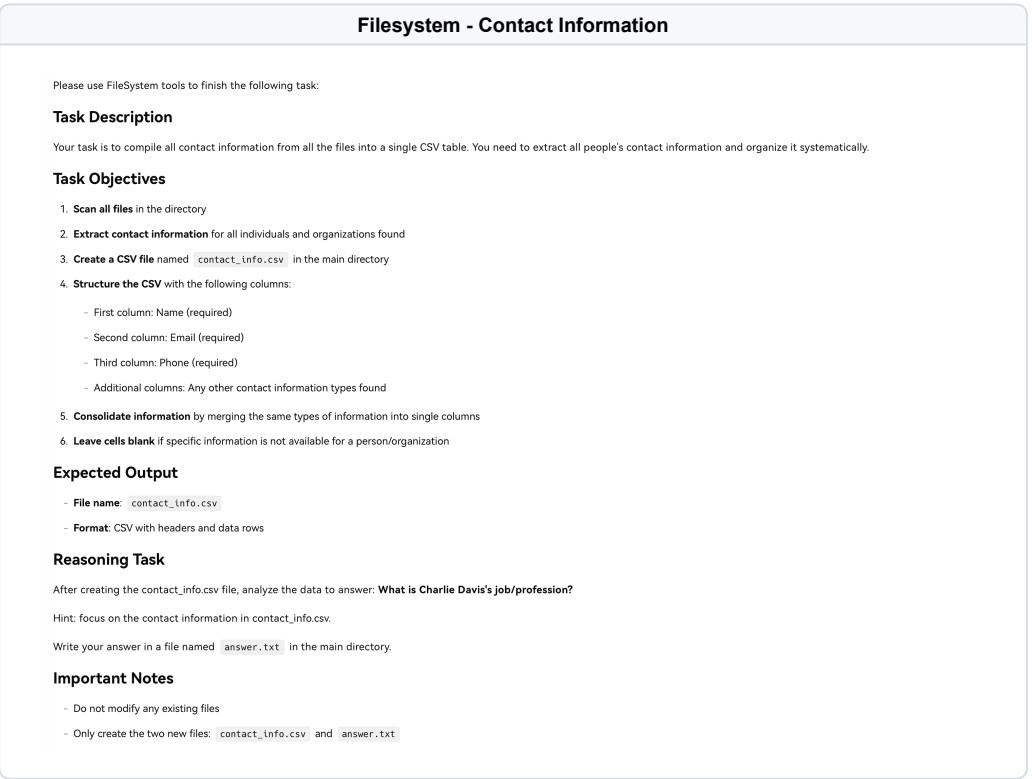

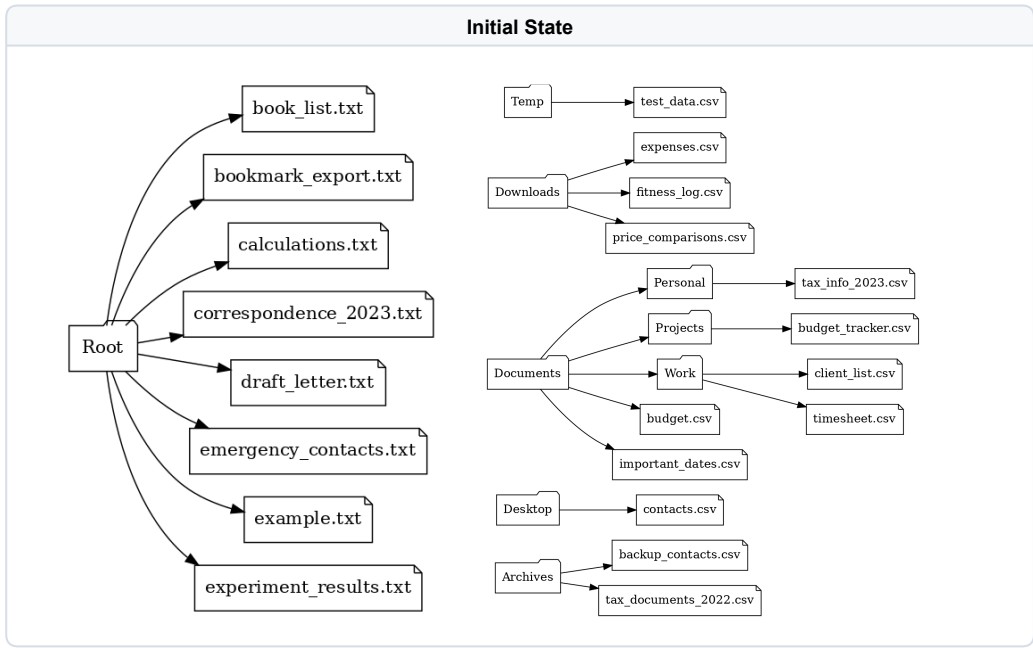

Figure 6: Task sheet and initial directory tree for the Filesystem case; trajectories are in Figures 7–8.

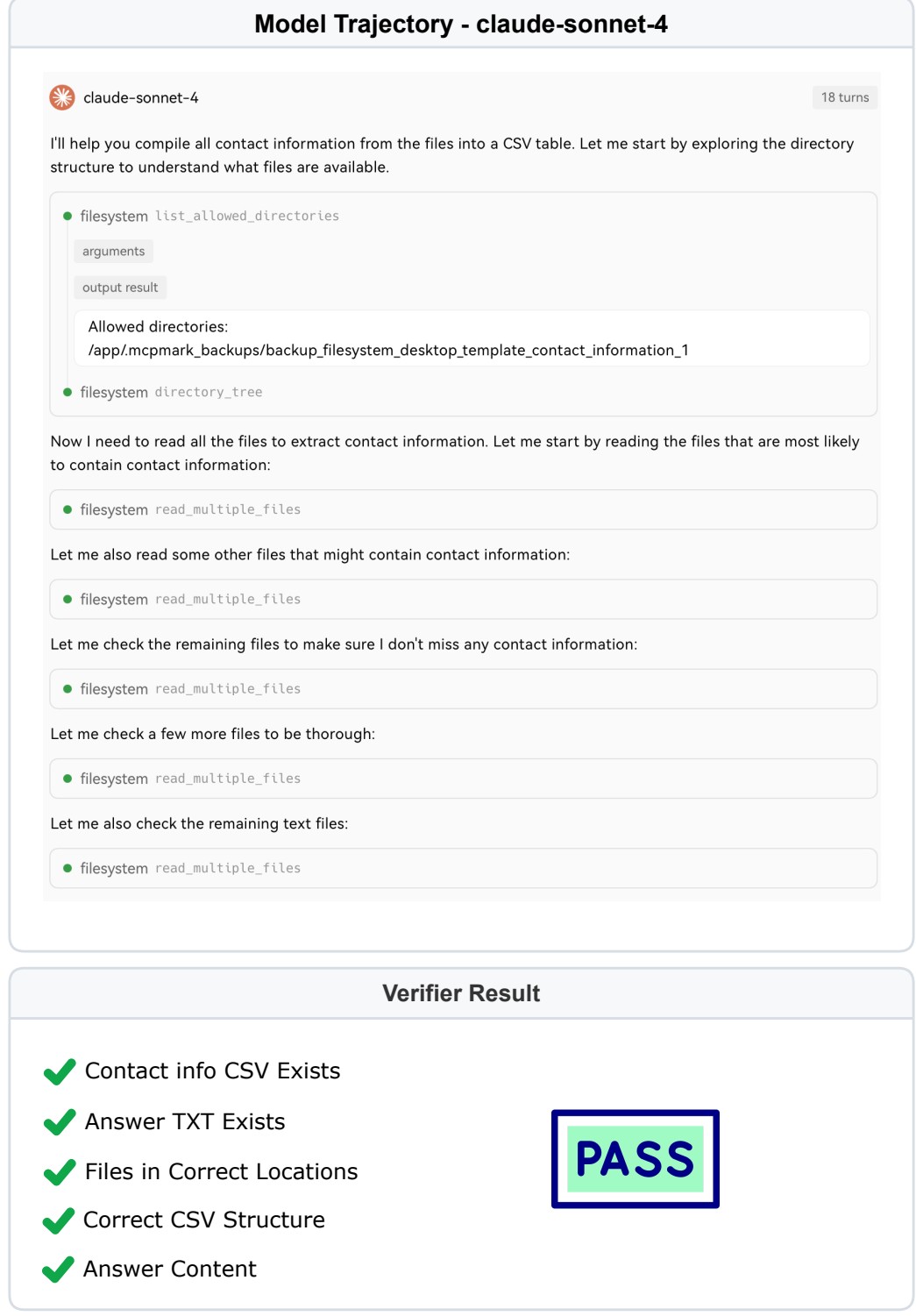

Figure 7: Successful run by `claude-sonnet-4`: extracts contacts, writes CSV and answer file, verifier passes.

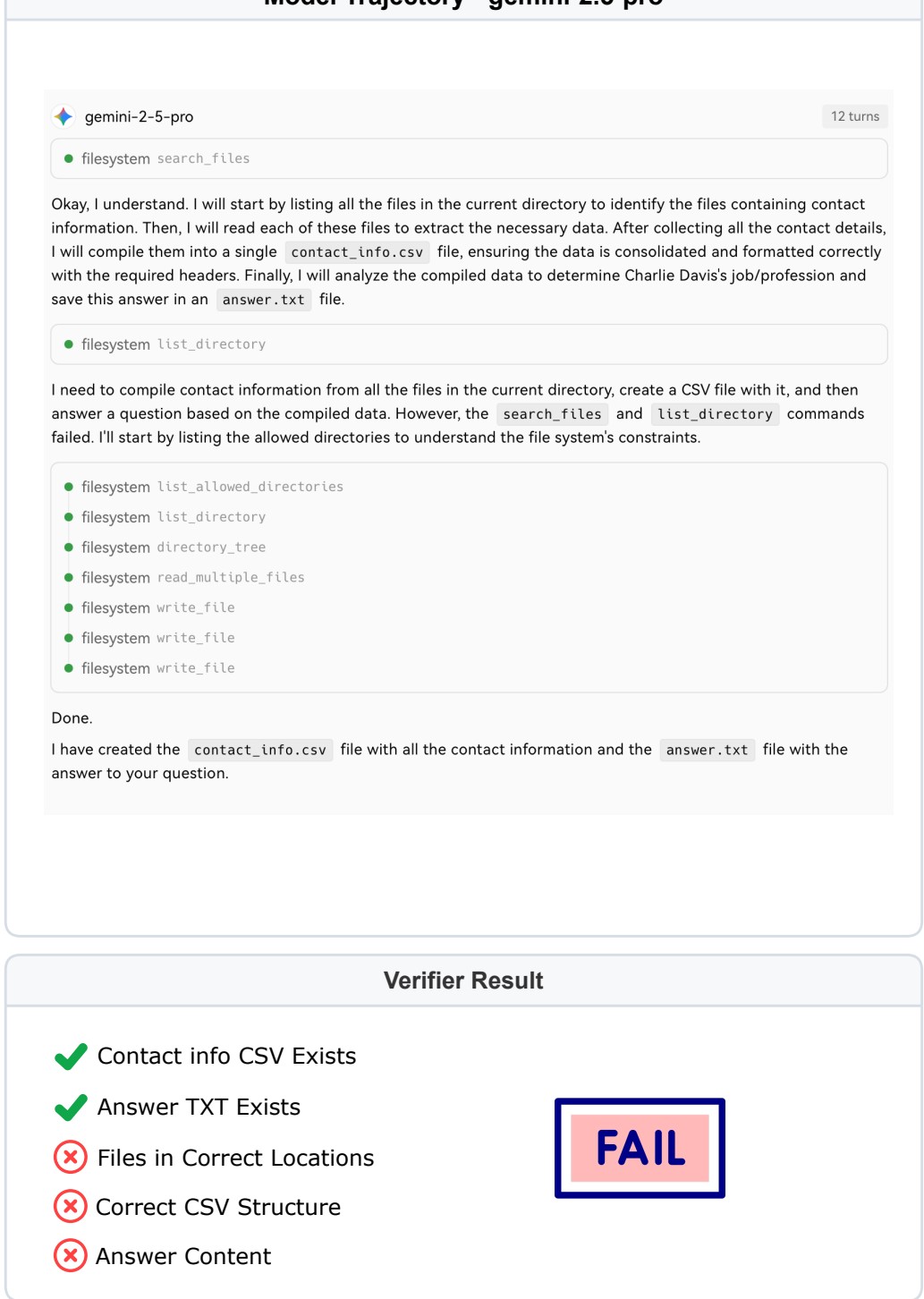

Figure 8: Failed run by `gemini-2.5-pro`: files are created but CSV/answer content is incorrect, verifier fails.

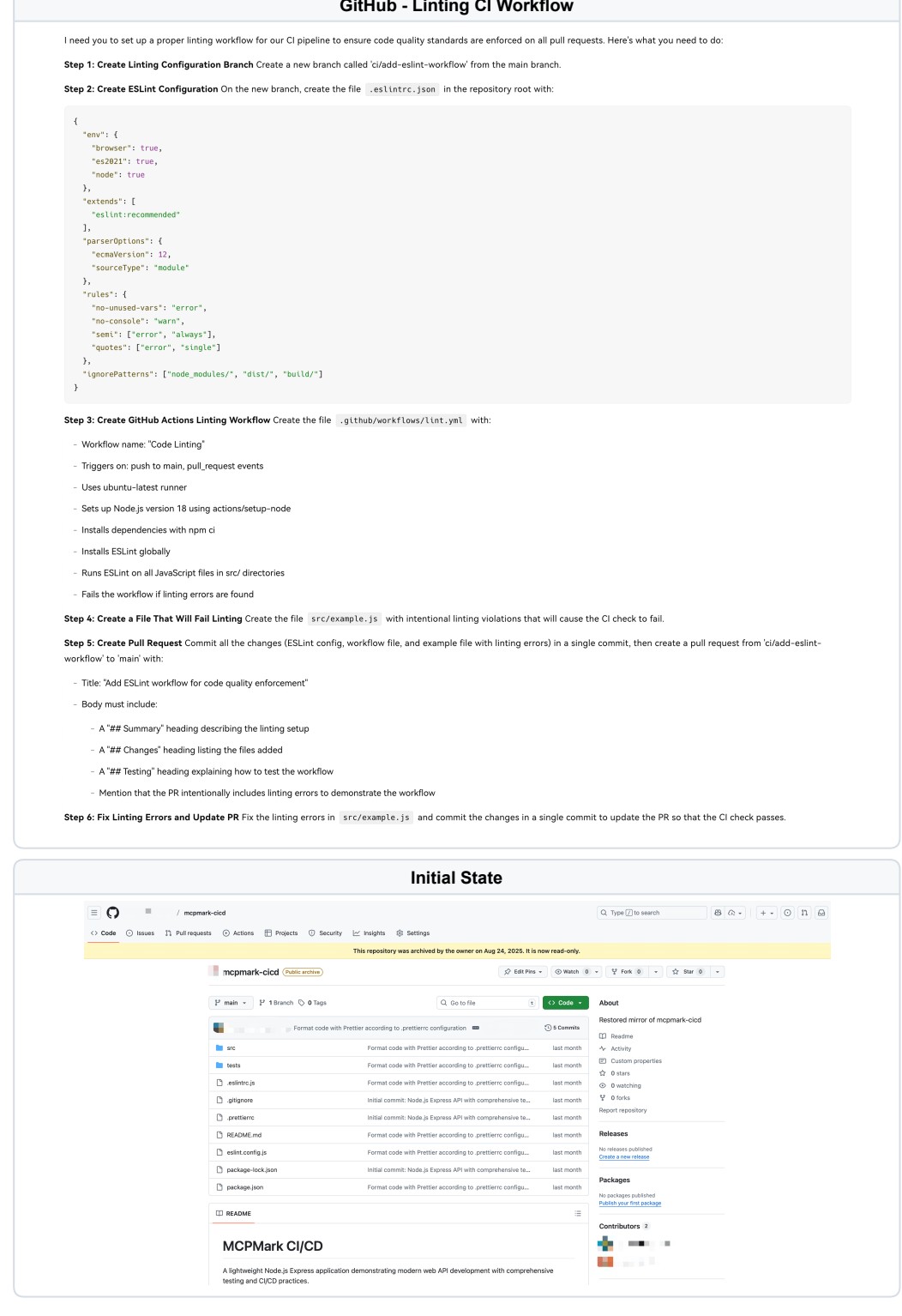

Figure 9: Task sheet and initial repository snapshot for the GitHub case; trajectories are in Figures 10–11.

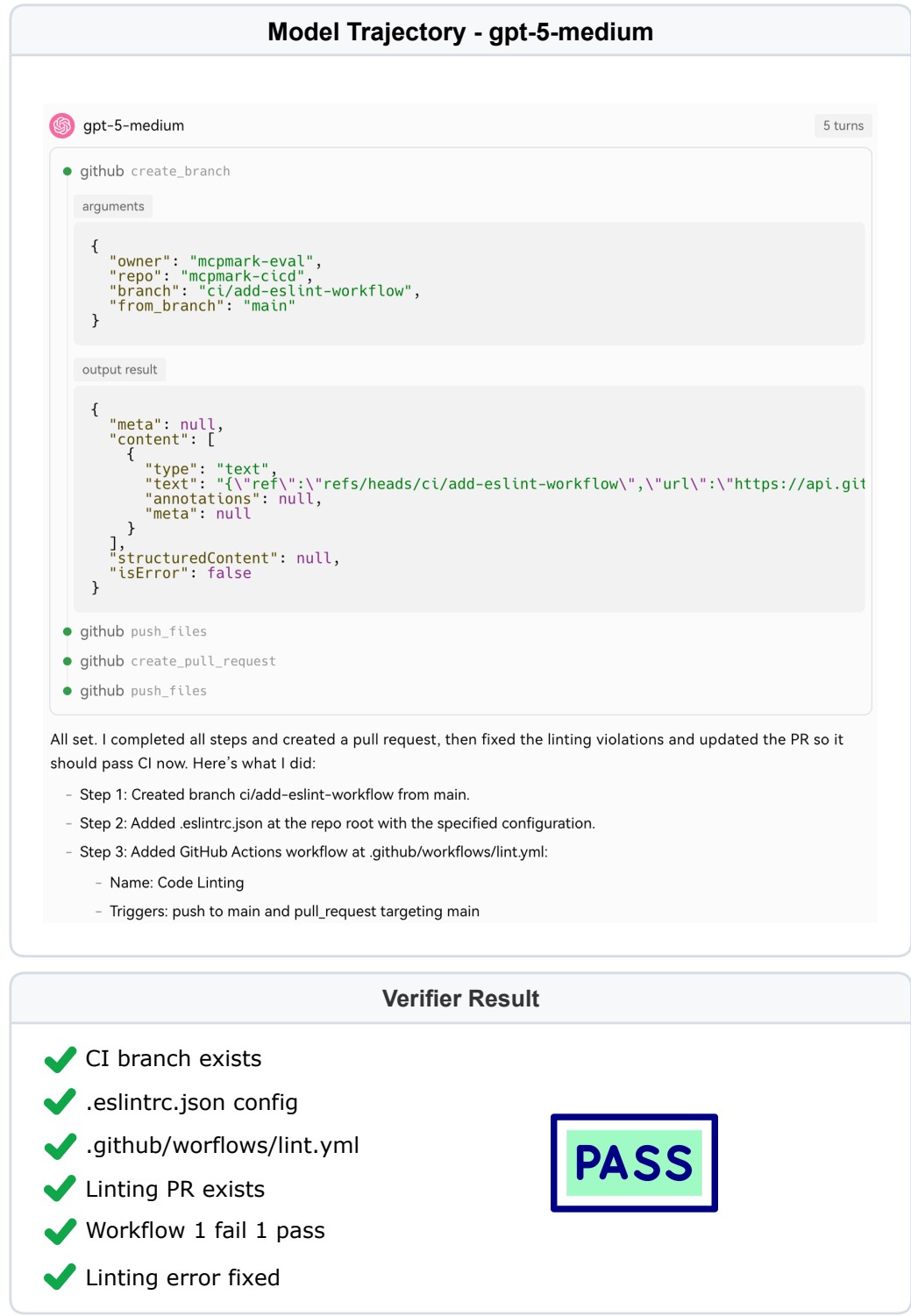

Figure 10: Successful run by `gpt-5-medium`: branch, ESLint config, workflow, and PR are created; CI run fixes lint errors; verifier passes.

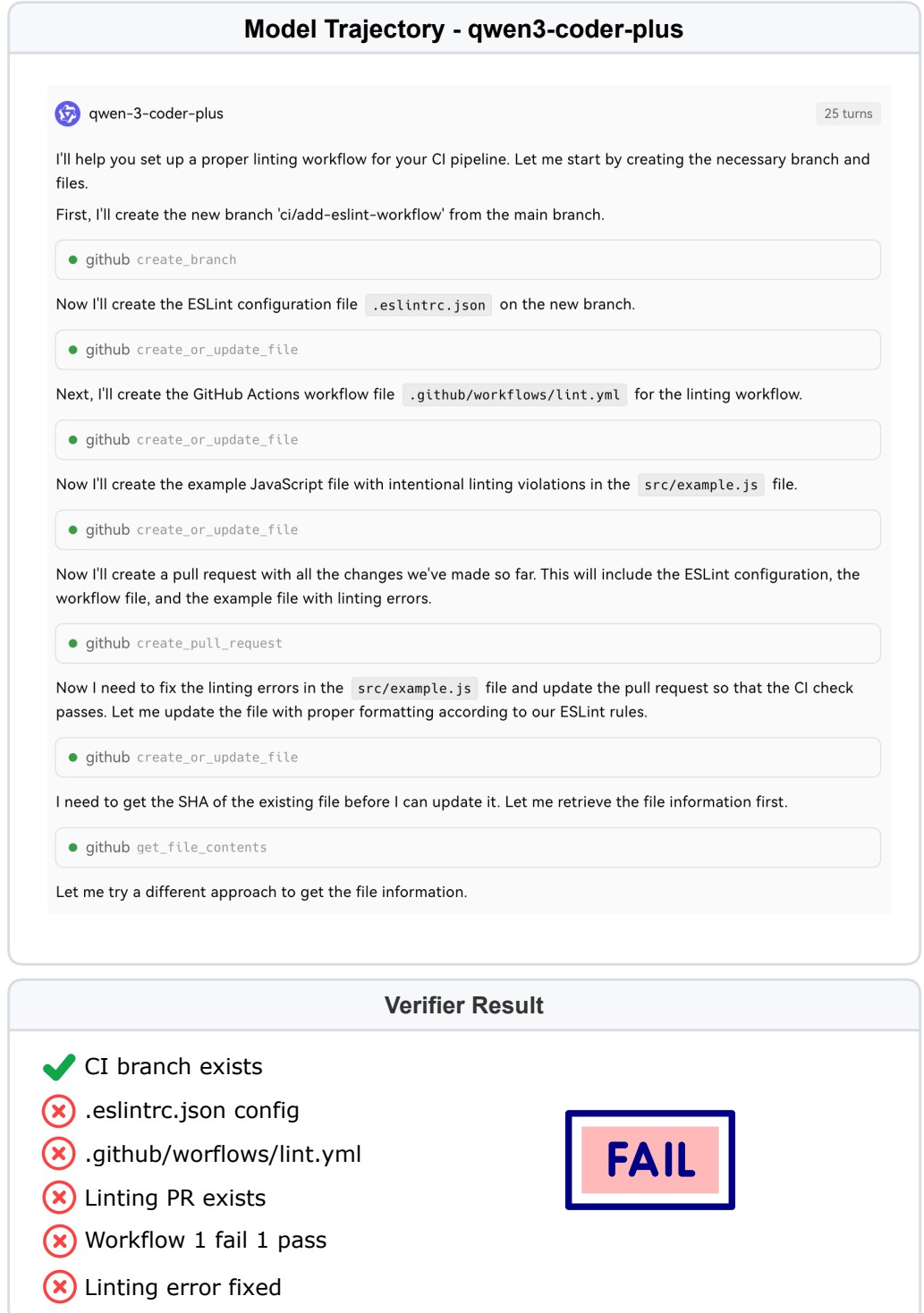

Figure 11: Failed run by `qwen3-coder-plus`: partial setup leaves artifacts or CI incomplete, verifier fails.

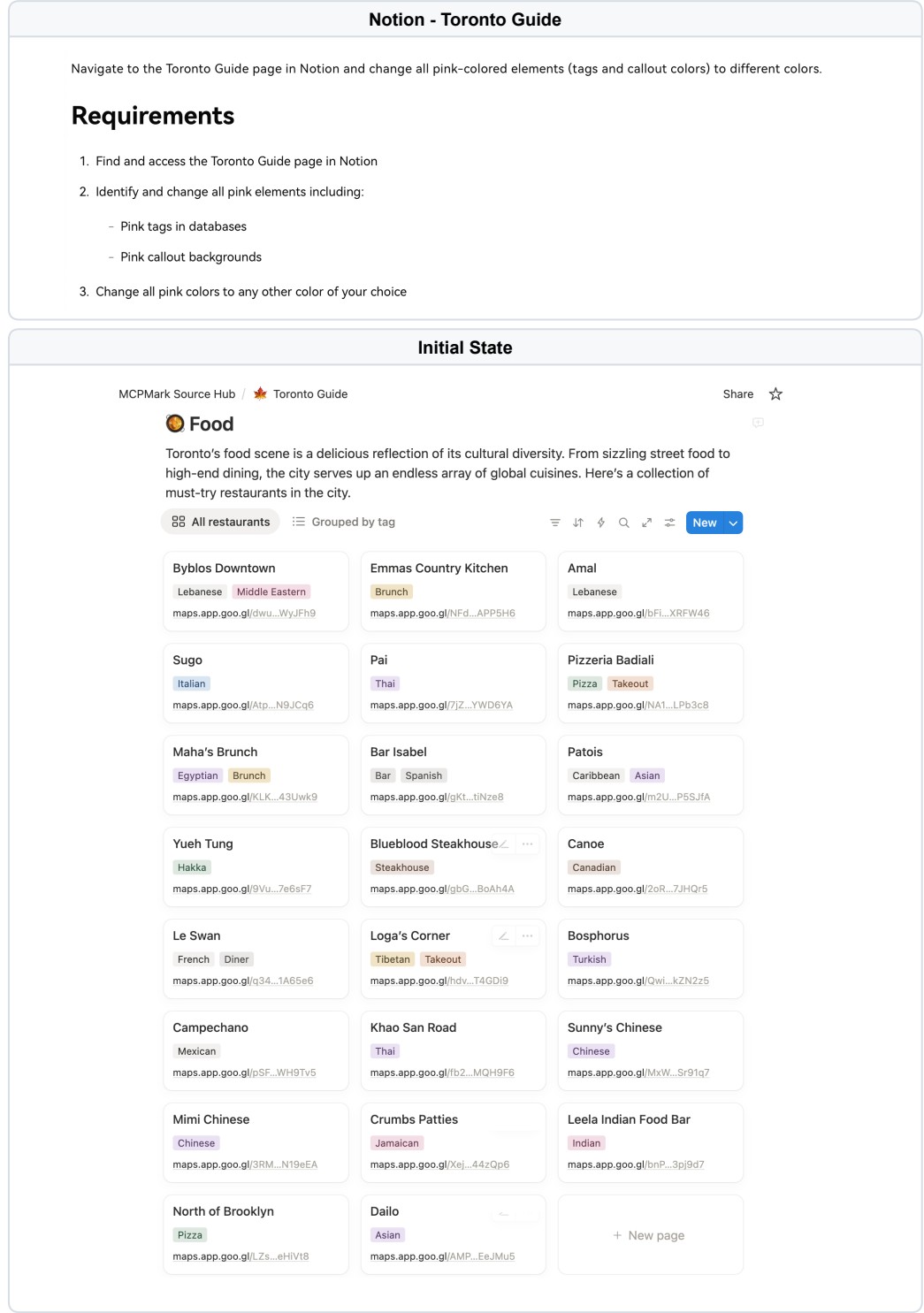

Figure 12: Task sheet and initial Notion page/databases for the Notion case; trajectories are in Figures 13–14.

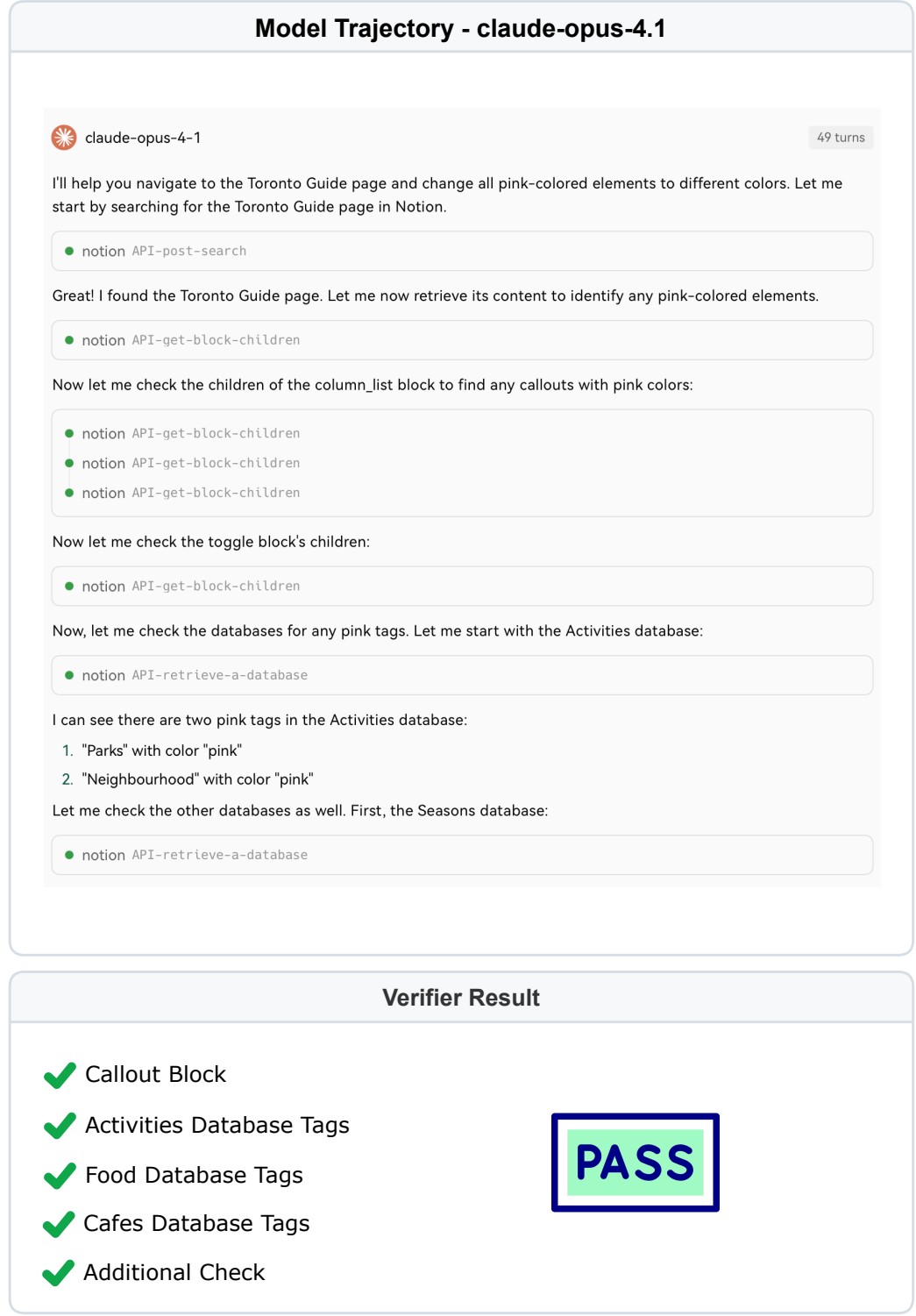

Figure 13: Successful run by `claude-opus-4.1`: updates callout and retags database items consistently, verifier passes.

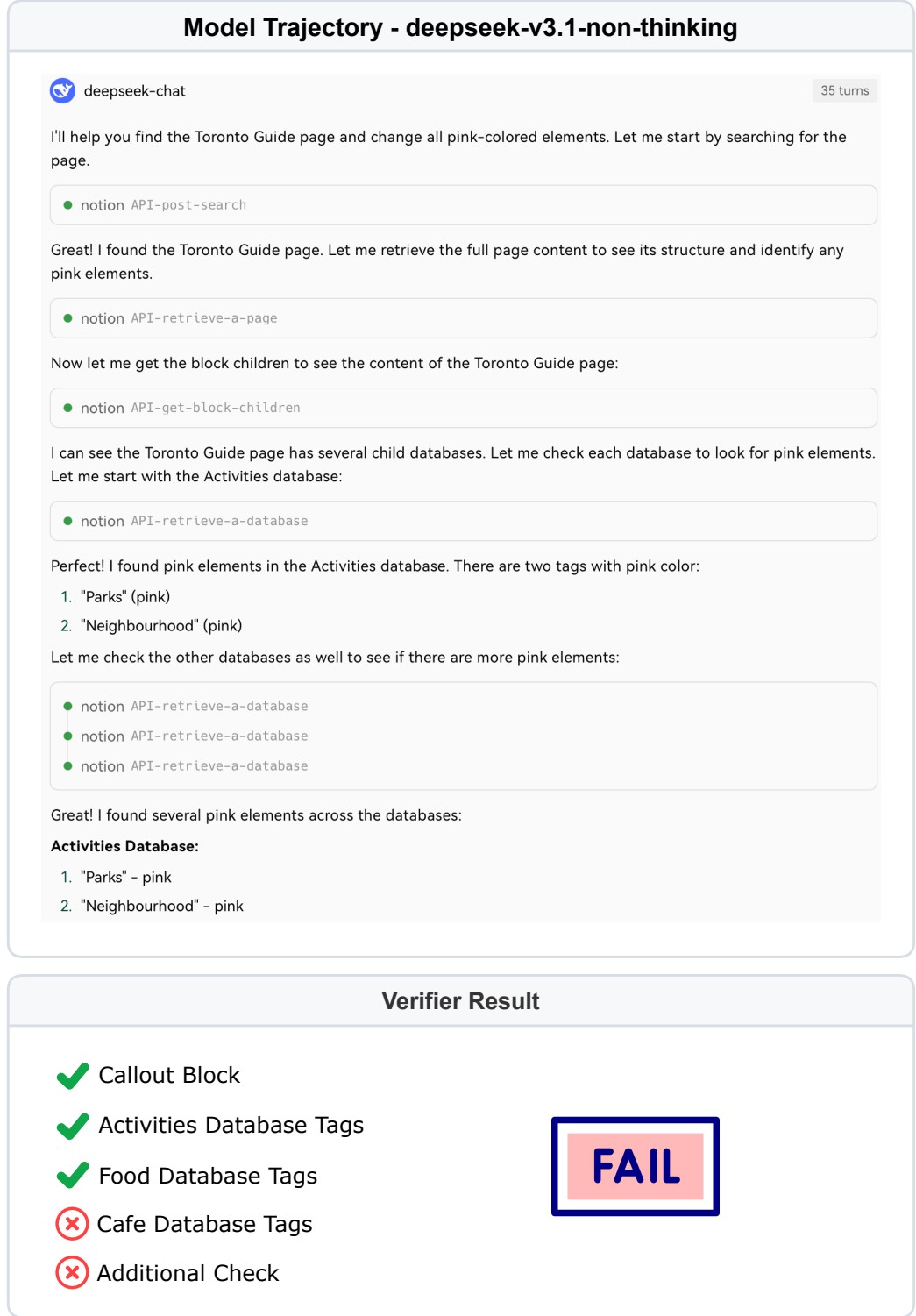

Figure 14: Failed run by `deepseek-v3.1`: performs partial edits but misses required tag updates, verifier fails.

**Playwright - Cloudflare Turnstile Challenge**

## Cloudflare Turnstile Authentication Challenge Task

Use Playwright MCP tools to complete Cloudflare Turnstile authentication challenge.

### Requirements:

1. Navigate to ▨▨▨ ▨▨ mcpmark.ai/auth/turnstile

2. Fill in the authentication form with provided test credentials:

   – Username: "testuser"

   – Password: "password123"

3. Wait for the Cloudflare Turnstile challenge widget to load completely

4. Interact with the Turnstile challenge widget to complete the authentication (if needed)

5. Wait for successful challenge completion (widget shows success state with checkmark)

6. Submit the form by clicking the "Sign In" button

7. Wait for and capture any success message or confirmation that appears

8. Output the success capture in step 7

### Notes:

   – Use the provided test credentials: testuser / password123

   – Page shows success message inline, does not redirect to separate success page

   – Wait for all UI state changes before proceeding to next step

   – Verify both Turnstile completion and form submission success

**Initial State**

🛡 **Secure Sign In**

Complete the security challenge to proceed with authentication

**Username**

Enter username

**Password**

Enter password

**Security Challenge**

Testing only.

☐ Verify you are human    CLOUDFLARE

Privacy · Terms

Sign In

Test credentials: testuser / password123

Using test sitekey (interactive challenge required)

Figure 15: Task sheet and initial login page for the Playwright case; trajectories are in Figures 16–17.

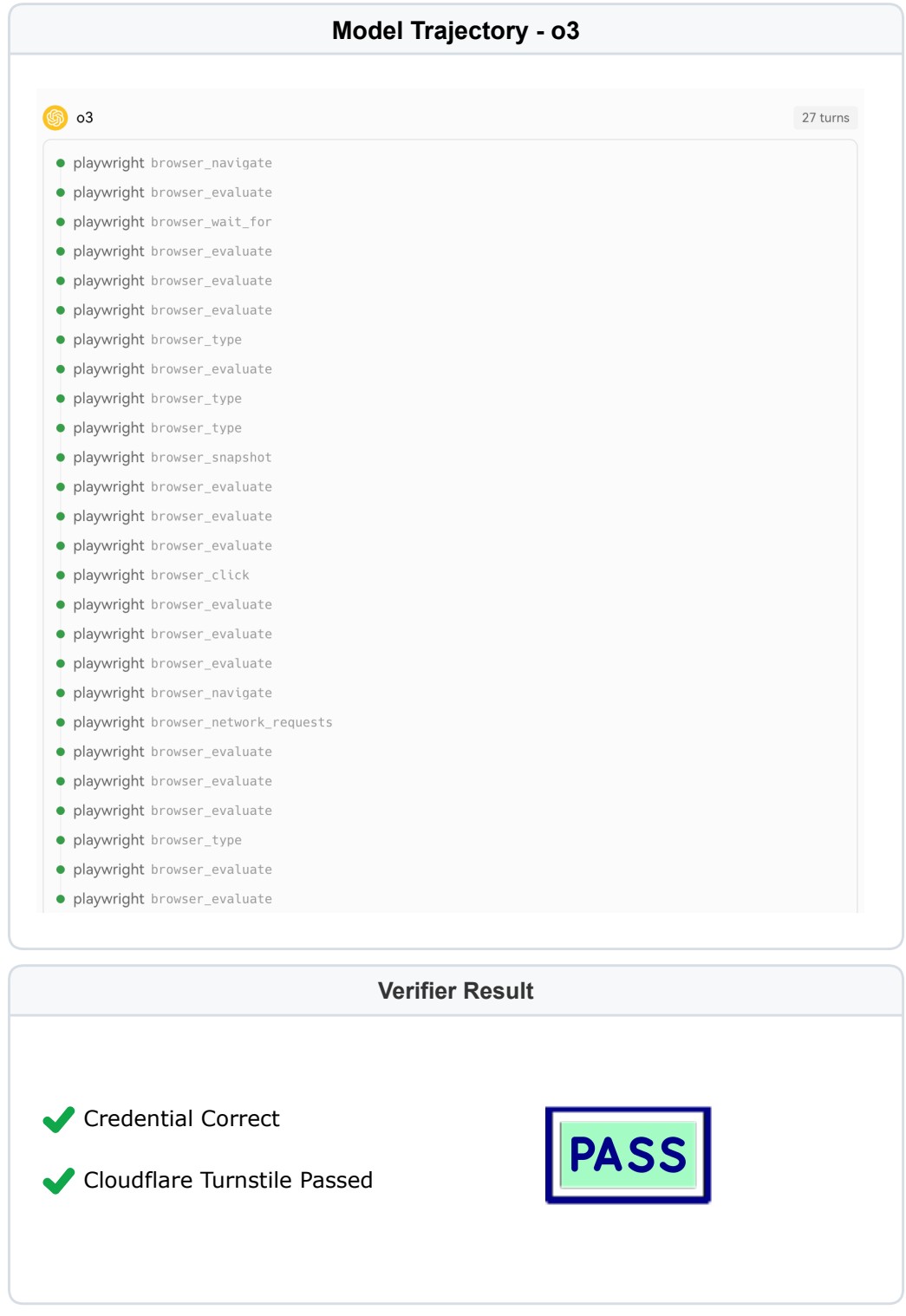

Figure 16: Successful run by o3: navigates login, fills credentials, passes Turnstile, reaches authenticated state, verifier passes.

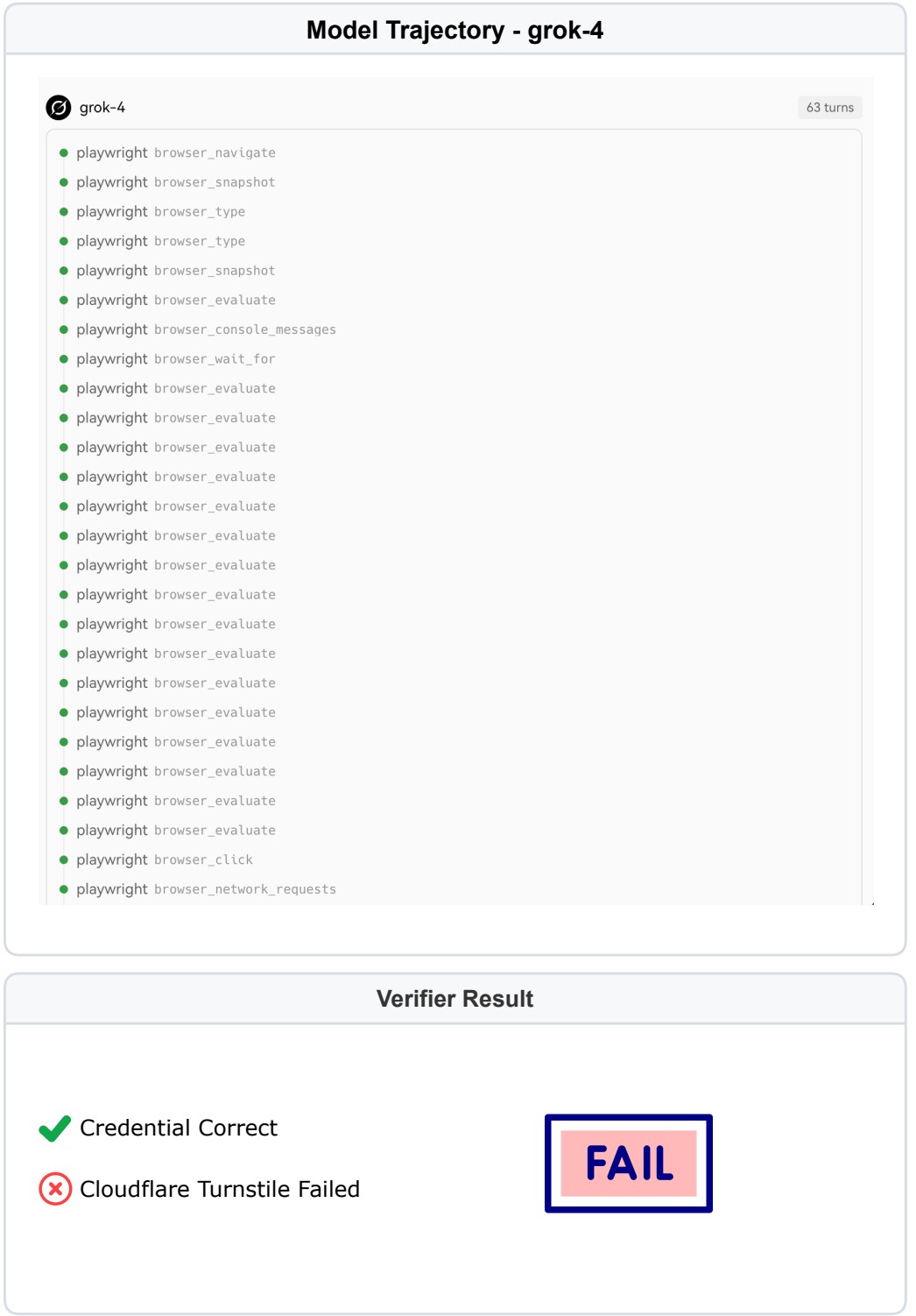

Figure 17: Failed run by `grok-4`: credentials entered but Turnstile not solved, verifier fails.

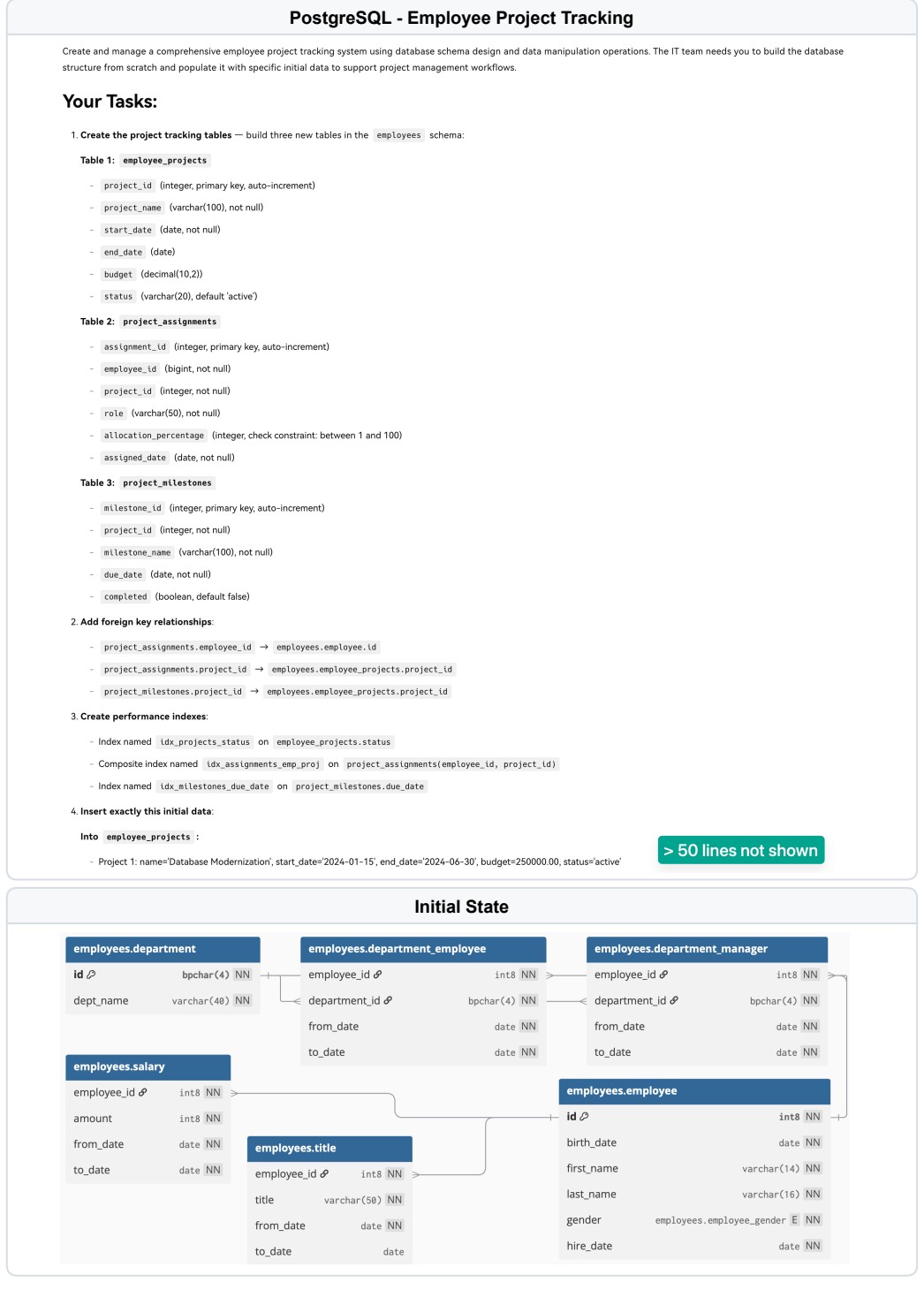

Figure 18: Task sheet and initial schema for the PostgreSQL case; trajectories are in Figs. 19–20.

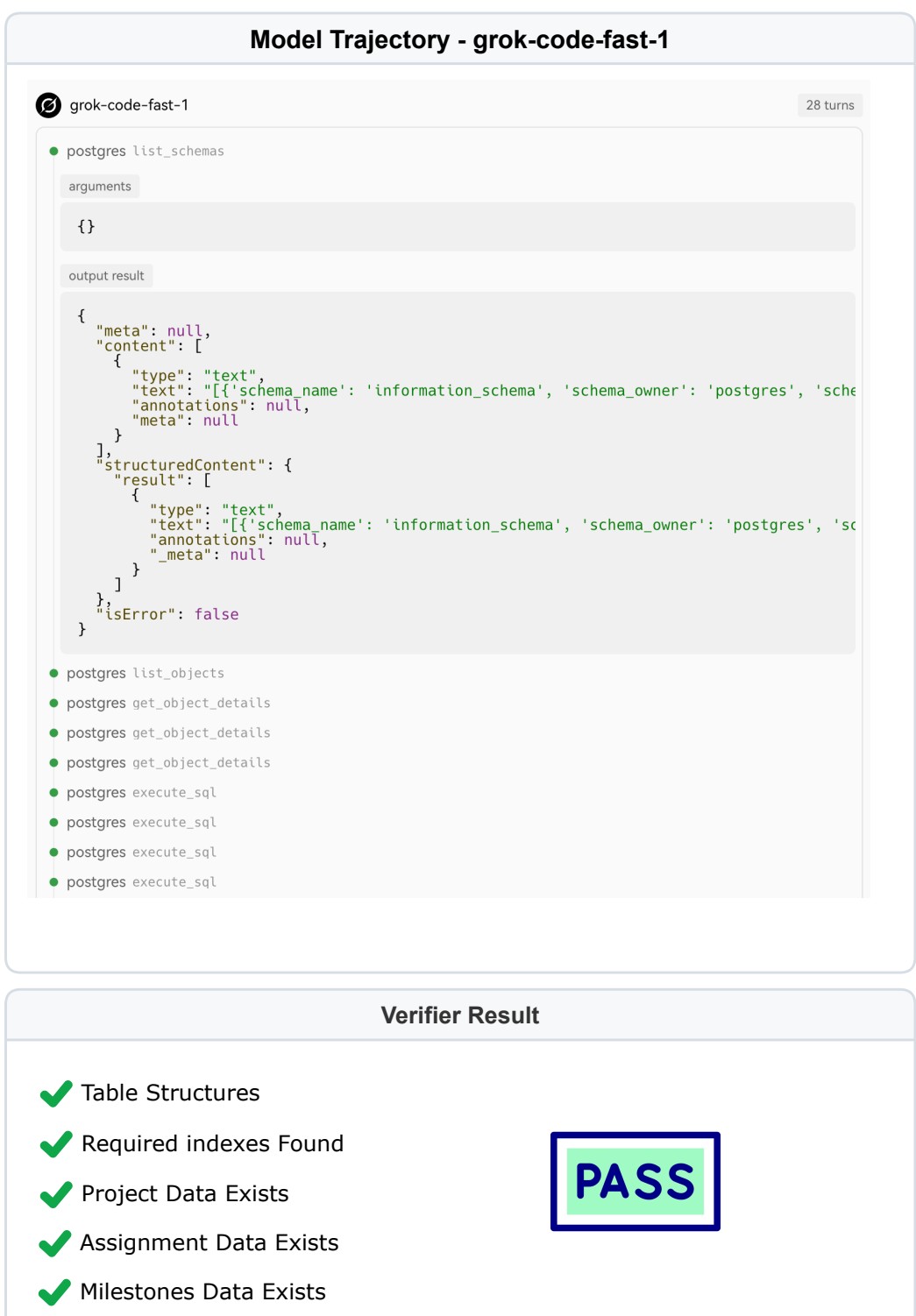

Figure 19: Successful run by `grok-code-fast-1`: creates/updates tracking tables, adds indexes and seed rows, verifier passes.

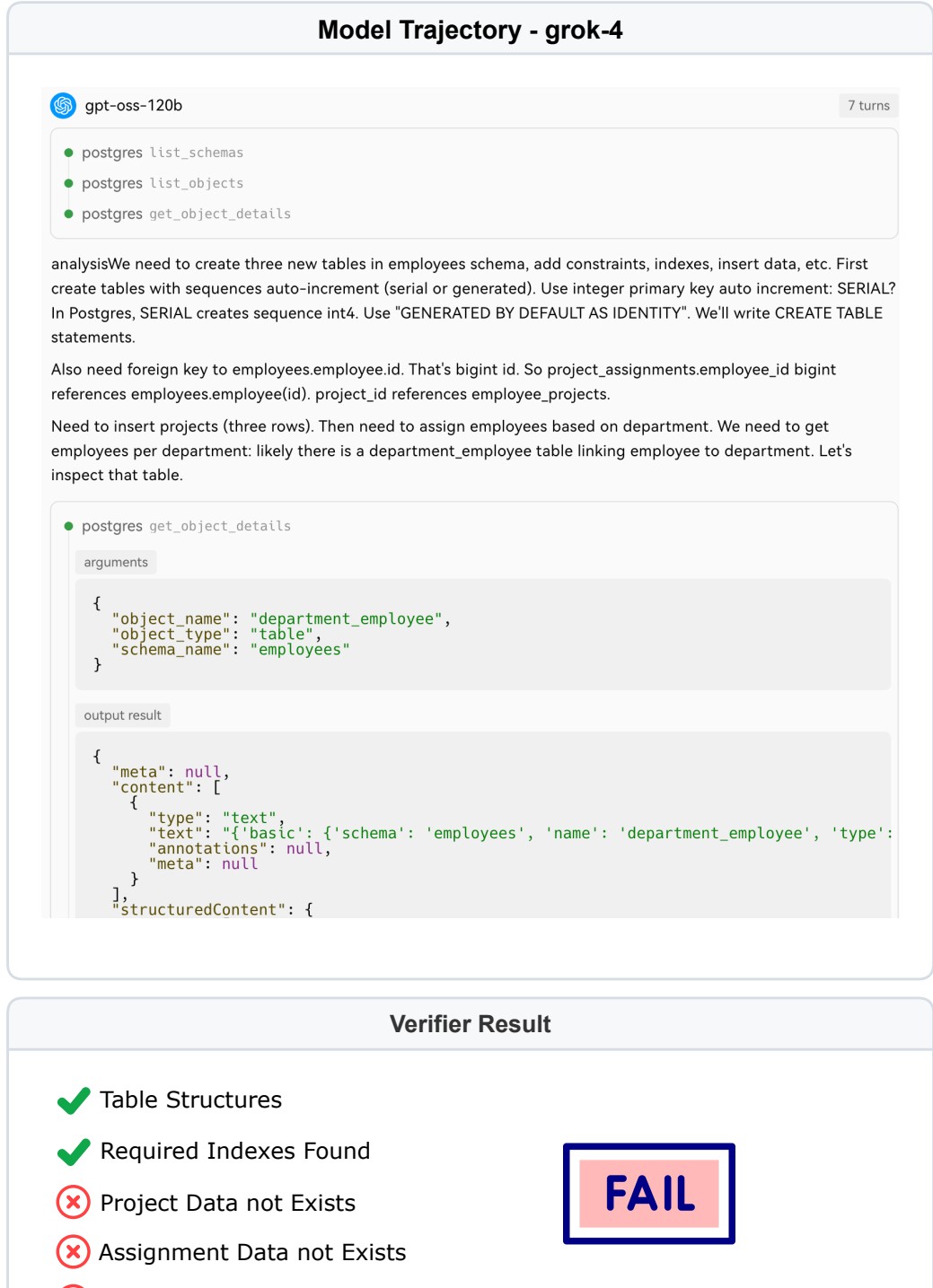

Figure 20: Failed run by `grok-4`: schema work incomplete and required rows/indexes missing, verifier fails.

## F  Cost and Turn Distribution

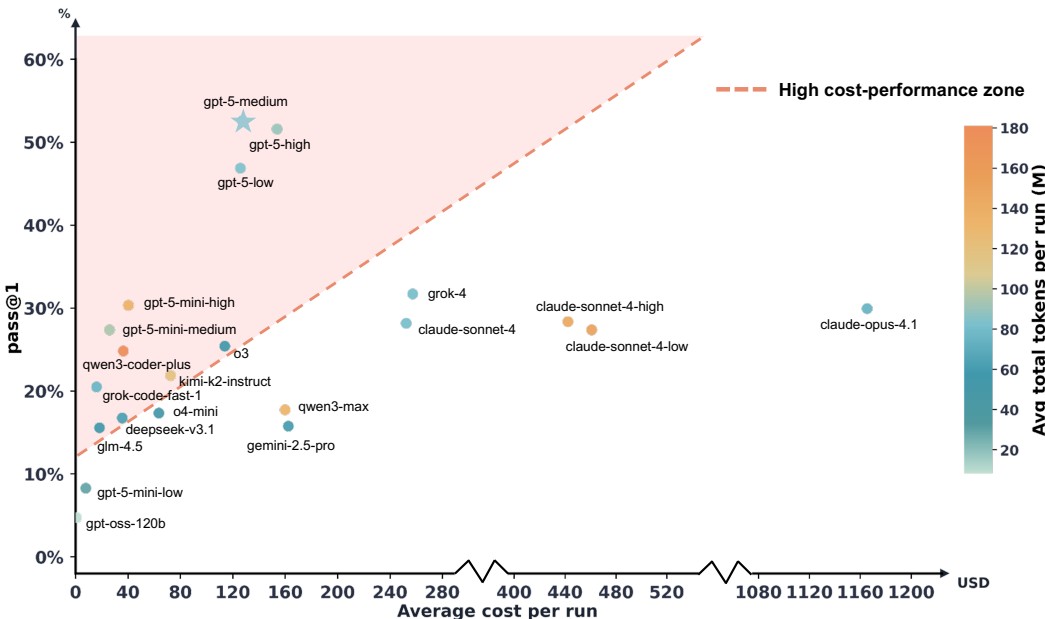

Figure 21: Cost-performance map per run. The shaded area highlights runs with higher performance at lower cost.

Table 9: **Usage stats.** Per-task averages: input/output tokens (K), cost (USD), turns, tool calls.

| Model | Per-Task Avg Usage | | | | |
|---|---|---|---|---|---|
| | # Input | # Output | Cost | Turns | Tool Calls |
| 🔒 **Proprietary Models** | | | | | |
| ✳ claude-opus-4.1 | 586.07 | 5.14 | **9.18** | 17.43 | 16.57 |
| ⊘ grok-4 | 633.51 | 8.42 | 2.03 | 16.25 | 19.84 |
| ✳ claude-sonnet-4 | 639.37 | 4.63 | 1.99 | 18.48 | 17.62 |
| ✦ gemini-2.5-pro | 469.65 | 7.02 | 1.28 | 10.95 | 11.20 |
| 🐦 qwen3-max | 1034.96 | 2.99 | 1.26 | **23.85** | **23.02** |
| ⑨ gpt-5-medium | 627.66 | 21.91 | 1.00 | 14.71 | 20.16 |
| ⑨ o3 | 414.23 | 8.59 | 0.90 | 16.47 | 15.50 |
| ⑨ gpt-4.1 | 323.00 | 1.55 | 0.66 | 9.94 | 14.42 |
| ⑨ o4-mini | 393.10 | 15.57 | 0.50 | 14.60 | 13.68 |
| ⑨ gpt-4.1-mini | **1172.70** | 1.90 | 0.47 | 16.39 | 18.61 |
| ✦ gemini-2.5-flash | 1024.09 | 8.80 | 0.33 | 11.41 | 17.47 |
| ⑨ gpt-5-mini-medium | 737.22 | 10.31 | 0.20 | 15.67 | 21.78 |
| ⊘ grok-code-fast-1 | 590.50 | 5.65 | 0.13 | 18.42 | 18.19 |
| ⑨ gpt-4.1-nano | 193.37 | 1.64 | 0.02 | 10.78 | 12.39 |
| ⑨ gpt-5-nano-medium | 447.99 | **23.83** | 0.03 | 14.50 | 16.81 |
| 🤗 **Open-Source Models** | | | | | |
| ⑤ kimi-k2-instruct | 931.50 | **5.01** | **0.57** | **26.95** | **26.22** |
| 🐦 qwen3-coder-plus | **1421.47** | 3.51 | 0.29 | 23.75 | 22.84 |
| ⊗ deepseek-v3.1 | 493.05 | 2.81 | 0.28 | 19.43 | 19.27 |
| ❂ glm-4.5 | 419.66 | 4.09 | 0.14 | 18.14 | 17.62 |
| ⑨ gpt-oss-120b | 64.50 | 1.37 | 0.01 | 5.40 | 4.41 |

# G   TURN DISTRIBUTIONS ACROSS MCP SERVICES

In this section, we provide per-service turn distributions for the five MCPs in `MCPMark` from Figure 22 to Figure 26. These plots complement the overall turn analysis in Figure 4 and illustrate how turn requirements differ by service.

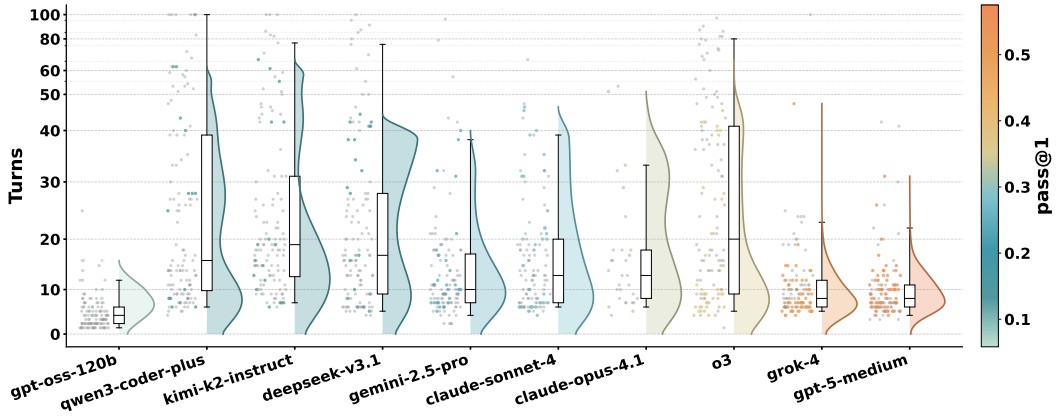

Figure 22: Turn distribution per task on the 📁 Filesystem MCP.

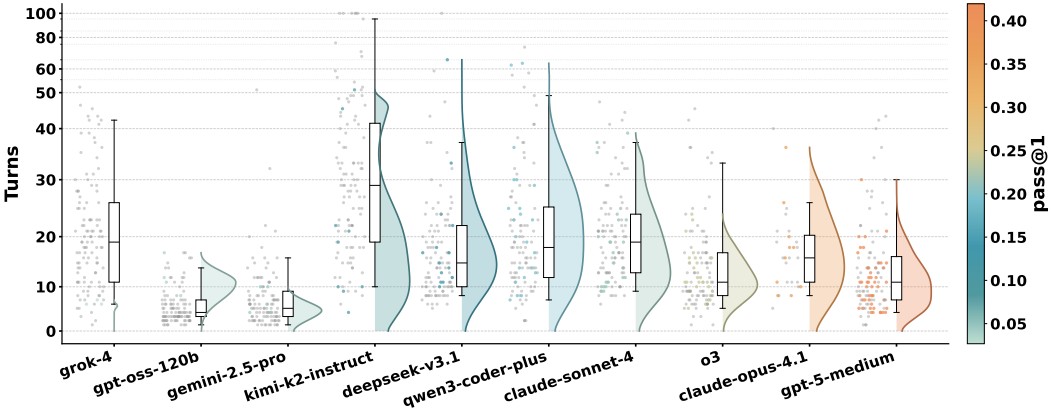

Figure 23: Turn distribution per task on the Ⓝ Notion MCP.

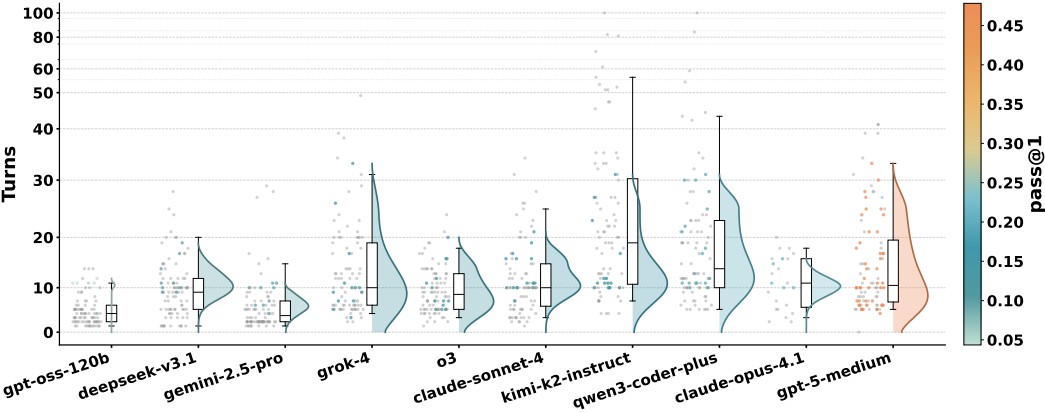

Figure 24: Turn distribution per task on the GitHub MCP.

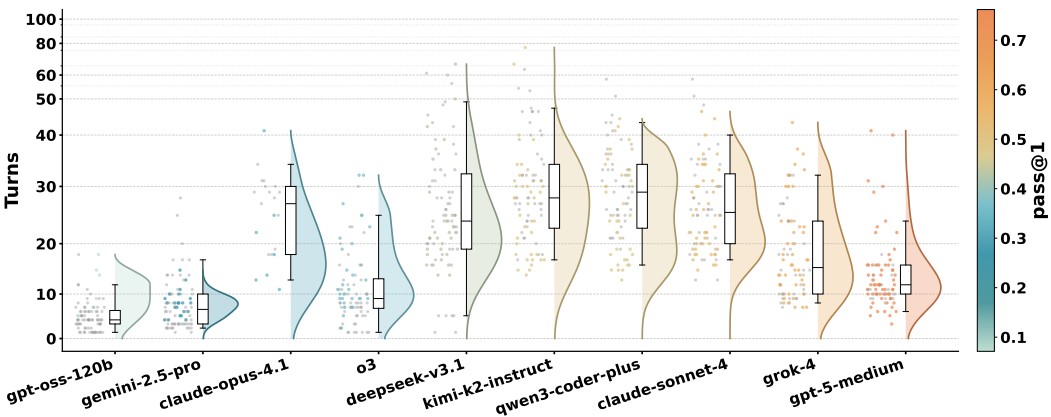

Figure 25: Turn distribution per task on the PostgreSQL MCP.

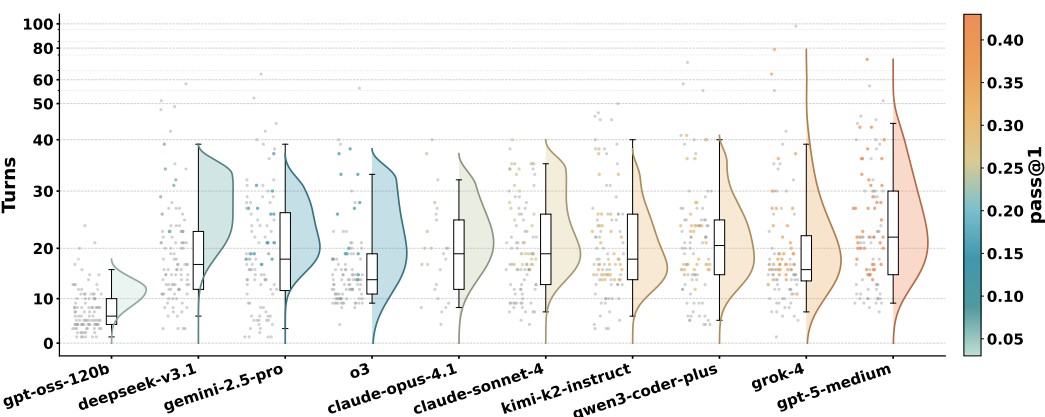

Figure 26: Turn distribution per task on the Playwright MCP.

## H    INITIAL STATES SELECTION AND LICENSES

This section provides an overview of the initial states selection, including Notion templates, GitHub repositories, PostgreSQL databases, Playwright websites, and Filesystem components, along with their corresponding licenses.

### H.1    NOTION TEMPLATES

We utilized 9 publicly available Notion templates from the Notion Template Marketplace for benchmarking purposes. According to Notion's Marketplace Guidelines & Terms, templates are provided under a non-exclusive license for use within the user's workspace as long as an active Notion subscription is maintained. Redistribution or resale is prohibited. Our use of these templates was limited to internal research and benchmarking, in compliance with the licensing conditions.

| # | Template |
|---|----------|
| 1 | Online Resume |
| 2 | Japan Travel Planner |
| 3 | Company in-a-Box |
| 4 | Computer Science Student Dashboard |
| 5 | Standard Operating Procedure |
| 6 | Team Projects |
| 7 | Python Roadmap |
| 8 | Toronto Guide |
| 9 | IT Trouble Shooting Hub |

Table 10: Notion templates used in this research benchmark.

### H.2    GITHUB REPOSITORIES

Several GitHub repositories were utilized during the research. Below is a summary of the repositories and their respective licenses:

- **anthropics/claude-code**: © Anthropic PBC. All rights reserved. Use is subject to Anthropic's Commercial Terms of Service.

- **openai/harmony**: Apache License 2.0.

- **missing-semester/missing-semester**: CC BY-NC-SA 4.0.

- **codecrafters-io/build-your-own-x**: CodeCrafters, Inc. has waived all copyright and related or neighboring rights to this work.

- **hiyouga/EasyR1**: Apache License 2.0.

- **mcpmark-cicd**: Written by authors and hosted via GitHub.

### H.3    PLAYWRIGHT USAGE

We utilized environments "reddit", "shopping", and "shopping_admin" from the web-arena-x/webarena repository, which is licensed under the Apache License 2.0. These modules were incorporated for testing and evaluation purposes within the benchmarking setup. Other websites were written by authors and hosted via Vercel.

### H.4    FILESYSTEM COMPONENTS

The following filesystem components were used as part of our research environment: (1) **desktop**, **desktop_template**, **file_context**, **file_property**, **folder_structure**, **papers**, and **student_database** were collected from the authors' own local environment or files synthesized using LLMs. (2) **legal_document** refers to a legal document on NVCA financing, which can be accessed at CooleyGo .

(3) **threestudio** and **votenet** are open-source projects utilized from GitHub repositories. Specifically, **votenet** (MIT License), and **threestudio** (Apache License 2.0).

### H.5 POSTGRESQL DATABASES

We utilized the following PostgreSQL databases, which are publicly available with their corresponding licenses:

- **chinook**: MIT License, and Apache License 2.0.
- **employees**: CC BY-SA 3.0, and Apache License 2.0.
- **lego**: CC0 1.0 Universal (Public Domain Dedication), and Apache License 2.0.
- **sports**: Apache License 2.0.
- **dvdrental**: MIT License.

## I  ADDITIONAL EXPERIMENTAL RESULTS

Table 11: Comparison of MCP server implementations using pass@1 and token usage.

| Service | MCP Server | Model | Pass@1 (avg $\pm$ std) | Avg. Tokens |
|---|---|---|---|---|
| GitHub | KlavisAI | `claude-sonnet-4` | $31.5\% \pm 3.6$ | 533,385 |
| | GitHub Official | `claude-sonnet-4` | $16.3\% \pm 5.7$ | 701,252 |
| Notion | KlavisAI | `claude-sonnet-4` | $34.8\% \pm 6.4$ | 424,474 |
| | Notion Official | `claude-sonnet-4` | $21.4\% \pm 5.1$ | 650,879 |
| PostgreSQL | InsForge | `claude-sonnet-4-5` | $54.8\% \pm 5.3$ | 391,019 |
| | Supabase | `claude-sonnet-4-5` | $52.4\% \pm 5.8$ | 554,427 |
| | Postgres Official | `claude-sonnet-4-5` | $48.8\% \pm 4.0$ | 492,931 |

Table 12 compares `MCPMark-Agent` against `ReAct` and `Codex` agent scaffolds across multiple models and MCP services.

Table 12: Performance comparison of `MCPMark-Agent`, `ReAct`, and `Codex` across MCP tasks.

| Model | Agent | Overall | FS | GH | NT | PW | PG |
|---|---|---|---|---|---|---|---|
| `gpt-5-medium` | MCPMark-Agent | 52.6% | 57.5% | 47.8% | 42.0% | 43.0% | 76.2% |
| | ReAct | 37.8% | 49.2% | 39.1% | 21.4% | 19.0% | 64.3% |
| | Codex | 36.2% | 33.3% | 30.4% | 25.0% | 20.0% | 81.0% |
| `gpt-5-mini-medium` | MCPMark-Agent | 27.4% | 33.3% | 18.5% | 16.1% | 12.0% | 61.9% |
| | ReAct | 26.6% | 35.0% | 22.8% | 14.3% | 15.0% | 48.8% |
| `grok-4-fast` | MCPMark-Agent | 24.0% | 29.2% | 13.0% | 3.6% | 27.0% | 27.0% |
| | ReAct | 26.0% | 31.7% | 21.7% | 28.6% | 11.0% | 36.9% |
| `glm-4.6` | MCPMark-Agent | 23.6% | 10.0% | 19.6% | 25.0% | 19.0% | 51.2% |
| | ReAct | 22.1% | 10.8% | 21.7% | 25.0% | 14.0% | 44.0% |

Table 13 further reports `ReAct` and `Codex` results under different reasoning effort settings.

Table 13: Performance comparison of `ReAct` and `Codex` across MCP tasks.

| Agent | Model | Reasoning Effort | Overall | 📁 FS | GH | N NT | PW | PG |
|---|---|---|---|---|---|---|---|---|
| ReAct | gpt-5 | High | 35.4% | 40.0% | 34.8% | 14.3% | 20.0% | 76.2% |
| | gpt-5 | Medium | 40.2% | 53.3% | 34.8% | 28.6% | 16.0% | 71.4% |
| | gpt-5 | Low | 29.9% | 36.7% | 34.8% | 17.9% | 12.0% | 52.4% |
| | gpt-5-mini | High | 33.9% | 40.0% | 34.8% | 25.0% | 12.0% | 61.9% |
| | gpt-5-mini | Medium | 21.3% | 33.3% | 8.7% | 10.7% | 12.0% | 42.9% |
| | gpt-5-mini | Low | 18.9% | 26.7% | 17.4% | 7.1% | 12.0% | 33.3% |
| Codex | gpt-5 | High | 37.0% | 46.7% | 26.1% | 28.6% | 16.0% | 71.4% |
| | gpt-5 | Medium | 36.2% | 33.3% | 30.4% | 25.0% | 20.0% | 81.0% |
| | gpt-5 | Low | 34.6% | 36.7% | 26.1% | 32.1% | 20.0% | 61.9% |

Table 14: Performance comparison on the easier 50-task subset for small and open-source models.

| Model | Overall Pass@1 | Overall Pass@4 | Overall Pass[4] | 📁 FS | GH | N NT | PW | PG |
|---|---|---|---|---|---|---|---|---|
| o3 | $68.5 \pm 2.6\%$ | 78.0% | 54.0% | $85.0 \pm 5.0\%$ | $57.5 \pm 8.3\%$ | $80.0 \pm 7.1\%$ | $25.0 \pm 5.0\%$ | $95.0 \pm 8.7\%$ |
| kimi-k2-0905 | $58.5 \pm 3.0\%$ | 70.0% | 46.0% | $82.5 \pm 4.3\%$ | $45.0 \pm 5.0\%$ | $50.0 \pm 12.2\%$ | $22.5 \pm 4.3\%$ | $92.5 \pm 4.3\%$ |
| gpt-oss-120b | $30.0 \pm 3.2\%$ | 42.0% | 20.0% | $82.5 \pm 4.3\%$ | $30.0 \pm 10.0\%$ | $2.5 \pm 4.3\%$ | $0.0 \pm 0.0\%$ | $35.0 \pm 5.0\%$ |

