# OpenReview forum: "MCPMark: A Benchmark for Stress-Testing Realistic and Comprehensive MCP Use"
_ICLR.cc/2026/Conference — ICLR 2026 Poster_

### Official Review · Reviewer_NsiE · 2025-10-31

**Soundness:** 4
**Presentation:** 4
**Contribution:** 4
**Rating:** 10
**Confidence:** 3

**Summary:**

This paper introduces MCPMark, a benchmark designed to evaluate the realistic and comprehensive use of MCP. It includes 127 tasks across five MCP environments: Notion, GitHub, Filesystem, PostgreSQL, and Playwright. It uses a minimal agent that has a tool-calling/llm-calling loop to evaluate LLMs against these tasks. The authors evaluated popular models and found even the best performing model only achieves 52% at pass@1 and 33% at pass^4.

**Strengths:**

- The proposed benchmark contains realistic and diverse tools and tasks
- Each task includes automatic verification scripts.
- It uses a simple but standardized agent framework to just evaluate LLM tool calling capabilities
- It evaluates many different models and draw some interesting conclusions

**Weaknesses:**

None. I like the paper quite a lot.

**Questions:**

None. The paper is clearly written.

---

> ### Author Response · Authors · 2025-11-21
> **Thanks for the Strong Review: Continued Improvements to MCPMark**
>
> We thank the reviewer for the very positive comments and for recommending a strong accept. Below is a concise summary of the new results and updates, while keeping the original benchmark design and conclusions unchanged.
>
> ---
>
> # Overview
> MCPMark now supports benchmarking **models** (including using easy task for benchmarking small and open-source models), **MCP servers**, and **agent frameworks**, all enabled by its standardized MCP protocol design.
>
> ---
>
> ## 1. Benchmarking MCP server implementations
> Since all components speak MCP, we can directly substitute different servers implementing the same service. Our initial experiments show server design strongly affects final performance.
>
> | Service   | MCP server        | Model              | Pass@1 (avg ± std) |
> |-----------|-------------------|--------------------|--------------------|
> | GitHub    | KlavisAI          | claude-sonnet-4    | 31.5 ± 3.6%        |
> | GitHub    | Official GitHub   | claude-sonnet-4    | 16.3 ± 5.7%        |
> | Notion    | KlavisAI          | claude-sonnet-4    | 34.8 ± 6.4%        |
> | Notion    | Official Notion   | claude-sonnet-4    | 21.4 ± 5.1%        |
> | Postgres  | InsForge          | claude-sonnet-4-5  | 54.8 ± 5.3%        |
> | Postgres  | Supabase          | claude-sonnet-4-5  | 52.4 ± 5.8%        |
> | Postgres  | Official Postgres | claude-sonnet-4-5  | 48.8 ± 4.0%        |
>
> These comparisons highlight that MCPMark can directly evaluate **MCP server quality**, not only model capability.
>
> ---
>
> ## 2. Benchmarking agent frameworks
> Beyond MCPMark-Agent (a minimal loop), we now include ReAct-style and Codex-style agents.
>
> | Model | Agent | Overall | Filesystem | GitHub | Notion | Playwright | PostgreSQL |
> |-------|--------|---------|-----------|--------|--------|------------|------------|
> | gpt-5-medium | MCPMark-Agent | 52.6% | 57.5% | 47.8% | 42.0% | 43.0% | 76.2% |
> | | ReAct | 37.8% | 49.2% | 39.1% | 21.4% | 19.0% | 64.3% |
> | | Codex | 36.2% | 33.3% | 30.4% | 25.0% | 20.0% | 81.0% |
> | gpt-5-mini-medium | MCPMark-Agent | 27.4% | 33.3% | 18.5% | 16.1% | 12.0% | 61.9% |
> | | ReAct | 26.6% | 35.0% | 22.8% | 14.3% | 15.0% | 48.8% |
> | grok-4-fast | MCPMark-Agent | 24.0% | 29.2% | 13.0% | 3.6% | 27.0% | 27.0% |
> | | ReAct | 26.0% | 31.7% | 21.7% | 28.6% | 11.0% | 36.9% |
> | glm-4.6 | MCPMark-Agent | 23.6% | 10.0% | 19.6% | 25.0% | 19.0% | 51.2% |
> | | ReAct | 22.1% | 10.8% | 21.7% | 25.0% | 14.0% | 44.0% |
>
> These results show that MCPMark naturally exposes differences between agent frameworks under realistic MCP workloads.
>
> ---
>
> ## 3. Easier 50-task subset for small and open-source models
> To better support smaller models, we released a 50-task “easy subset.” Tasks remain realistic but require fewer turns.
>
> | Model         | Overall Pass@1 | Overall Pass@4 | Overall Pass^4 | Filesystem | GitHub | Notion | Playwright | PostgreSQL |
> |---------------|-----------------|----------------|----------------|------------|--------|--------|------------|-------------|
> | o3            | 68.5 ± 2.6%     | 78.0%         | 54.0%         | 85.0 ± 5.0%  | 57.5 ± 8.3% | 80.0 ± 7.1% | 25.0 ± 5.0% | 95.0 ± 8.7% |
> | kimi-k2-0905  | 58.5 ± 3.0%     | 70.0%         | 46.0%         | 82.5 ± 4.3%  | 45.0 ± 5.0% | 50.0 ± 12.2% | 22.5 ± 4.3% | 92.5 ± 4.3% |
> | gpt-oss-120b  | 30.0 ± 3.2%     | 42.0%         | 20.0%         | 82.5 ± 4.3%  | 30.0 ± 10.0% | 2.5 ± 4.3%  | 0.0 ± 0.0%  | 35.0 ± 5.0% |
>
> This new subset enables meaningful evaluation of small and OSS models while preserving realistic complexity patterns.
>
> ---
>
> ## Final Remarks
> We sincerely thank the reviewer for the encouraging review, as well as the exceptionally positive rating. We deeply appreciate the reviewer’s careful reading and recognition of our design choices. We hope that the additional experiments further reinforce the clarity and reliability of our contributions, and we would be delighted if these results help raise the reviewer’s confidence in our work.
>
> We also promise our commitment to the community: **MCPMark will continue to be actively maintained, upgraded, and used as a shared benchmark**.

---

### Official Review · Reviewer_VEYJ · 2025-10-31

**Soundness:** 3
**Presentation:** 2
**Contribution:** 3
**Rating:** 6
**Confidence:** 2

**Summary:**

The paper proposes a new benchmark MCPMark for evaluating more relaistic use of Model Context Protocol (MCP) in context of LLM agents. Model Context protocol (MCP) standarizes how LLMs connect with external systems and tools. However, the paper argues that existing MCP bencmarks are often too narrow or not realistic. The proposed benchmark proposes to stress-test MCP use across 127 tasks in more realistic environments and larger interaction depth. The paper provides some interesting analysis on implicit vs explicit errors for state-of-the-art LLMs on MCPMark. Finally, they also show that best LLMs often require much more interactions (tool calls and execution turns) than those required by previous benchmarks.

**Strengths:**

* The paper highlights an interesting problem of evaluating realistic use of MCP in context of LLM agents.

* The analysis on implicit vs explicit errors and failure cases in Sec. 4.2 is interesting.

* The paper shows results on state-of-the-art LLMs on MCPMark (Tab. 3), showing the need for better stress testing for MCP use.

**Weaknesses:**

* Can the authors please provide details on the scalability of the current data curation pipeline?
       *  As mentioned in Sec. 2.1 and L164, the data curation pipeline requires a human expert. Thus, it will be important to also discuss the scalability of the data curation pipeline.

* Also in Sec. 2.1, the authors mention that for each task "including computer science PhD students, front-end designers, full-stack & AI infra engineers, and AI investors—each task takes 3~5 hours of focused expert effort". While not a major concern, providing more details on impact of human expertise on the task quality will be interesting.

* Which scaffold is used for MCPMark-Agent?

* Also can the authors have some intuition for effect of the used agent scaffold on the numbers reported in Tab. 3?

* Finally, in Sec. 4, in the discussion on failure cases it is noted that most errors are implicit. Could this be fixed by better prompting or more precise problem statements or does this point to a more fundamental limitation? If possible it will be interesting to see some qualitative examples of these implicit errors.

**Questions:**

* The authors mention in Sec. 3, that more turns or cost does not equal better performance. This is actually consistent with findings in SWE agents or coding tasks. Have the authors verified if the trend also holds with test-time scaling and parallel rollouts?

Please see the weaknesses section for some additional questions.

---

> ### Author Response · Authors · 2025-11-21
> **Response to Reviewer VEYJ [1/3]**
>
> Thank you for your supportive review and suggestions. Below we respond to the comments in **Weaknesses (W)** and **Questions (Q)**.
>
> > **W1: Can the authors please provide details on the scalability of the current data curation pipeline?**
>
> Thank you for highlighting this concern. MCPMark is designed as an evaluation benchmark, so our priority is to capture realistic, complex, and diverse MCP use. For this reason, we rely on human–AI cooperation during environment/context exploration and task creation. This approach increases fidelity but does limit scalability, which is acceptable for evaluation rather than large-scale instance creation.
>
> We agree that scalable data generation is important for training models, but that objective is outside the scope of MCPMark. That said, the exploration–then–synthesis structure in our pipeline offers a potential pathway toward scalable generation. Recent work such as WebExplorer [1] indicates that similar ideas can be extended in that direction.
>
> [1]. WebExplorer: Explore and Evolve for Training Long-Horizon Web Agents (arXiv 2025)
>
> > **W2: Providing more details on impact of human expertise on the task quality will be interesting.**
>
> That is a good and interesting question. The impact of our team's diverse expertise is most significant during the **initial task ideation and scoping phase**. This is when we brainstorm the main workflow topics for a given MCP scenario. In general, our team's multi-disciplinary expertise allows us to source high-quality, domain-specific tasks grounded in real-world challenges, not generic academic exercises.
>
> For example, some of our benchmark's most unique tasks originated from these specific backgrounds:
>
> - An AI investor contributed the automated browser login problem, reflecting a focus on user friction and onboarding workflows.
> - A full-stack engineer proposed a complex task to conduct a comprehensive security audit in a database, specifically to identify users with insufficient or dangling permissions.
> - A front-end engineer suggested the CI/CD pipeline problems, which test an agent's ability to manage code integration and deployment workflows.
>
> This diversity in expertise is what ensures the tasks are not just complex, but also grounded in practical, real-world utility.
>
> > **W3: Which scaffold is used for MCPMark-Agent?**
>
> Thank you for the question. MCPMark-Agent does not rely on any external agent scaffold. Instead, as described in Sec. 2.3, it is a lightweight evaluation framework that we implement directly on top of two components: LiteLLM and the official Model Context Protocol (MCP) Python SDK.
>
> - LiteLLM is used to convert MCP tools into the standardized function-call interface and to route model queries to each provider’s native API;
> - The MCP Python SDK is used to configure MCP servers and expose their tool specifications to the agent.
>
> This design keeps the agent minimal and avoids system-specific heuristics, making it suitable for standardized evaluation rather than production use.

---

> ### Author Response · Authors · 2025-11-21
> **Response to Reviewer VEYJ [2/3]**
>
> > **W4: Also can the authors have some intuition for effect of the used agent scaffold on the numbers reported in Tab. 3?**
>
> We provide additional experiments on different agent scaffolds. Although the patterns differ from the MCPMark-Agent baseline, the overall trend remains consistent with our manuscript: increasing reasoning effort generally leads to better performance, though the magnitude of improvement is still model- and MCP-dependent.
>
> | Agent         | Model      | Reasoning Effort | Overall | Filesystem | GitHub | Notion | Playwright | PostgreSQL |
> | ------------- | ---------- | ---------------- | ------- | ---------- | ------ | ------ | ---------- | ---------- |
> | MCPMark-Agent | gpt-5      | high             | 51.6%   | 52.5%      | 50.0%  | 44.6%  | 42.0%      | 72.6%      |
> |               | gpt-5      | medium           | 52.6%   | 57.5%      | 47.8%  | 42.0%  | 43.0%      | 76.2%      |
> |               | gpt-5      | low              | 46.9%   | 54.2%      | 27.2%  | 36.6%  | 45.0%      | 73.8%      |
> |               | gpt-5-mini | high             | 30.3%   | 35.0%      | 19.6%  | 20.5%  | 15.0%      | 66.7%      |
> |               | gpt-5-mini | medium           | 27.4%   | 33.3%      | 18.5%  | 16.1%  | 12.0%      | 61.9%      |
> |               | gpt-5-mini | low              | 8.3%    | 12.5%      | 8.7%   | 5.4%   | 1.0%       | 14.3%      |
> | ReAct         | gpt-5      | high             | 35.4%   | 40.0%      | 34.8%  | 14.3%  | 20.0%      | 76.2%      |
> |  | gpt-5      | medium           | 40.2%   | 53.3%      | 34.8%  | 28.6%  | 16.0%      | 71.4%      |
> | | gpt-5      | low              | 29.9%   | 36.7%      | 34.8%  | 17.9%  | 12.0%      | 52.4%      |
> | | gpt-5-mini | high             | 33.9%   | 40.0%      | 34.8%  | 25.0%  | 12.0%      | 61.9%      |
> | | gpt-5-mini | medium           | 21.3%   | 33.3%      | 8.7%   | 10.7%  | 12.0%      | 42.9%      |
> | | gpt-5-mini | low              | 18.9%   | 26.7%      | 17.4%  | 7.1%   | 12.0%      | 33.3%      |
> | Codex | gpt-5 | high | 37.0%   | 46.7%      | 26.1%  | 28.6%  | 16.0%  | 71.4%  |
> | | gpt-5 | medium | 36.2%   | 33.3%      | 30.4%  | 25.0%  | 20.0%  | 81.0%  |
> | | gpt-5 | low | 34.6%   | 36.7%      | 26.1%  | 32.1%  | 20.0%  | 61.9%  |
>
> > **W5.1: Finally, in Sec. 4, in the discussion on failure cases it is noted that most errors are implicit. Could this be fixed by better prompting or more precise problem statements or does this point to a more fundamental limitation?**
>
> Thank you for the question. In our setting, the task descriptions are already validated during the task-creation stage to remove ambiguity, so the implicit failures we observe do not come from unclear prompts. They reflect the model’s core limits in reasoning, planning, tool use, and handling long contexts, and our analysis indicates that most errors arise from these fundamental capability limits.
>
> We also include results from other agent scaffolds in the table below. As shown, Codex and ReAct, although designed as general-purpose scaffolds, do not surpass the naive iterative tool-calling framework (i.e., MCPMark-Agent). This suggests that improving model performance on MCP tasks through prompting or scaffold adjustments alone is not straightforward and not trivial. More targeted and carefully designed scaffolds may be needed to obtain real improvements, which we see as a promising direction for future work.
>
> > **W5.2: It will be interesting to see some qualitative examples of these implicit errors.**
>
> Regarding the qualitative examples of these implicit errors, you can first check the provided examples in Appendix E. Since it is difficult to include the entire trajectory and evaluation feedback in the paper, we have uploaded the trajectories of each model on each task. You can feel free to review them in the supplementary materials. Specifically, for each task folder, you can check `messages.json` to see the raw execution trajectory and `meta.json` to view all relevant metadata, including error messages.

---

> ### Author Response · Authors · 2025-11-21
> **Response to Reviewer VEYJ [3/3]**
>
> > **Q1: Have the authors verified if the trend (i.e., more turns or cost does not equal better) also holds with test-time scaling and parallel rollouts?**
>
> **On Test-Time Scaling (Reasoning Effort)**. Section 4.1 directly evaluates models under different reasoning-effort settings. This allows us to study test-time scaling because increasing reasoning effort produces more thinking tokens and thus higher inference-time cost. The results show that additional test-time cost does not reliably improve MCP performance. Instead, the effect depends strongly on both the model architecture and the task environment.
>
> - **Model-Dependent Capacity**: The benefits of increased reasoning varied significantly by model. For instance, `gpt-5-mini`'s performance scaled dramatically (8.27% to 30.32% pass@1), demonstrating a clear ability to leverage additional reasoning. Conversely, models like `gpt-5-nano` and `claude-sonnet-4` were "insensitive" to added effort, suggesting they lack the requisite capacity to translate additional reasoning into effective action.
> - **MCP Service-Dependent Environment**: Furthermore, the performance gains were highly contingent on the MCP service environment. The most substantial improvements occurred on remote services (e.g., GitHub, where `gpt-5` improved from 27.17% to 50.00%). Local services (e.g., PostgreSQL) saw negligible changes. This suggests that increased reasoning is most beneficial for tasks requiring stronger generalization, such as those involving less-frequently (in training data) encountered remote APIs.
>
> Overall, these findings support our claim that “more turns or more cost” does not guarantee better agentic behavior.
>
> **On Parallel Rollouts.** We did not include parallel rollouts in MCPMark. The evaluation agent is intentionally minimal so that we can isolate each model’s *intrinsic* abilities under a standard MCP tool-calling loop. Although parallel rollouts can improve performance—as demonstrated in recent systems such as Kimi-K2-Thinking (heavy)—these techniques are computationally expensive and would obscure the baseline capabilities we aim to measure.
>
> That said, parallel inference strategies are a promising direction for future studies. They may especially benefit **smaller models**, where the gap between capability and cost-efficiency is most relevant and where test-time compute allocation can meaningfully shift performance.
>
> Thank you again for your thoughtful review. We hope these clarifications address your concerns, and we kindly ask you to consider raising your score if the clarifications are helpful.

---

### Official Review · Reviewer_ZaUC · 2025-11-02

**Soundness:** 3
**Presentation:** 3
**Contribution:** 3
**Rating:** 6
**Confidence:** 4

**Summary:**

The paper adds to the collection of MCP benchmarks. The key distinguishing aspects of this benchmark relative to existing MCP benchmarks are the following:
- It introduces more complex tasks requiring larger number of turns
- It covers more operations (read/update/delete) relative to other benchmarks that are only read-heavy.

The benchmark ensures repeatable and reliable evaluation by setting up resettable initial states and the tasks are co-desgined by a human-LLM agent team, with a significant effort in reviewing and cross-reviewing the tasks to ensure quality control.

The evaluations illustrate the key challenges of todays LLM-based agents in addressing the tasks. The analysis provide useful categorization of the failure modes.

**Strengths:**

- The paper improves the existing MCP benchmarking space by contributing a more complex collection of tasks with rigorous programmatic evaluation.
- The benchmarking and analyses provide a useful empirical assessment of the state-of-the-art models (both closed and open) in tackling complex tasks using MCP servers.

**Weaknesses:**

I see two areas that can be improved:

- While I fully appreciate that creating these benchmarks take significant time and effort, the number of instances created overall and the pathway for creating more future instances dont appear to scale. Some justification of why 127 tasks are adequate (in a statistical sense) and representative (I do think has been argued for reasonably in the paper) needs to emphasized more clearly in the paper.

- The choice of a simple agent for comparative model evaluation is a valid argument. However, it would still be useful to know what is the best performance that at least one single model could have achieved given more sophisticated agent formulations would be useful to know. In practice, this is how agents are going to be deployed. In some sense, one would like to what is the best performance that can be achieved on this benchmark with the best known techniques today.

- The arguments for why MCP specific benchmarks are necessary can be stronger. There are multiple multi-tool use benchmarks that exist (e.g. AppWorld, WebArena, and SWEBench). For example, benchmarks like AppWorld provide a rich set of problems defined over multiple applications covering nine different applications and have tasks that are much longer in terms of turns than proposed in this paper. the Why shouldn't we consider MCP wrapping around these multi-tool and multi-turn agentic benchmarks? Why invest effort into creating new benchmarks.

**Questions:**

See weaknesses above.

---

> ### Author Response · Authors · 2025-11-21
> **Response to Reviewer ZaUC [1/2]**
>
> Thank you for your helpful and constructive review. Below we respond to the comments in **Weaknesses (W)**.
>
> > **W1.1: The number of instances and the task creation pathway dont appear to scale.**
>
> Thank you for raising this concern. MCPMark is intended as an **evaluation benchmark**, so our design prioritizes realism and faithfulness of MCP workflows rather than large-scale instance generation. As a result, our pipeline intentionally relies on **human–AI collaboration** during environment exploration and task creation, yielding high-fidelity tasks while inevitably limiting scalability. We view this as acceptable for evaluation benchmarks, which require depth and realism far more than size.
>
> We agree that scalable data generation is important for training models, but that objective is outside the scope of MCPMark. That said, the exploration–then–synthesis structure in our pipeline offers a potential pathway toward scalable generation. Recent work such as WebExplorer [1] indicates that similar ideas can be extended in that direction.
>
> In addition, to support benchmarking smaller and OSS models, we have released a **50-task easy subset**. These tasks still use realistic environments but require fewer branching steps, thus maintaining complexity patterns while being more accessible. This also partially alleviates the concern about scale. Preliminary results on this subset:
>
> | Model         | Overall Pass@1 | Overall Pass@4 | Overall Pass^4 | Filesystem | GitHub | Notion | Playwright | PostgreSQL |
> |---------------|-----------------|----------------|----------------|------------|--------|--------|------------|-------------|
> | o3            | 68.5 ± 2.6%     | 78.0%         | 54.0%         | 85.0 ± 5.0%  | 57.5 ± 8.3% | 80.0 ± 7.1% | 25.0 ± 5.0% | 95.0 ± 8.7% |
> | kimi-k2-0905  | 58.5 ± 3.0%     | 70.0%         | 46.0%         | 82.5 ± 4.3%  | 45.0 ± 5.0% | 50.0 ± 12.2% | 22.5 ± 4.3% | 92.5 ± 4.3% |
> | gpt-oss-120b  | 30.0 ± 3.2%     | 42.0%         | 20.0%         | 82.5 ± 4.3%  | 30.0 ± 10.0% | 2.5 ± 4.3%  | 0.0 ± 0.0%  | 35.0 ± 5.0% |
>
> MCPMark is open-sourced and fully versioned, and we promise to actively **expand, upgrade, and replace tasks using community contributions**. This addresses long-term scalability while keeping the current v1.0 task set consistent.
>
> [1]. WebExplorer: Explore and Evolve for Training Long-Horizon Web Agents (arXiv 2025)
>
> > **W1.2: Some justification of why 127 tasks are adequate (in a statistical sense) and representative needs to emphasized more clearly in the paper.**
>
> Thank you for the helpful comment. We agree that the justification for the adequacy and representativeness of the 127 tasks should be stated more explicitly, and we will revise the paper accordingly. Specifically:
>
> - For statistical adequacy, each model is evaluated on 127 tasks spanning 5 MCP services and 38 curated initial states, with 4 independent runs per task. This yields more than 500 trajectories per model per metric. As shown in Table 2, the standard deviation of pass@1 across runs is typically 1–3 percentage points, indicating that 127 tasks already provide stable aggregate estimates.
> - For representativeness, the tasks are constructed from realistic and diverse initial states, including widely used Notion templates and established GitHub repositories. Building on these, the tasks cover real MCP workflows that are absent from existing benchmarks, such as CI/CD triggering in GitHub, database relation and rollup configuration in Notion, authenticated browser automation in Playwright, PostgreSQL transactional operations, and multi-file organization in the Filesystem.

---

> ### Author Response · Authors · 2025-11-21
> **Response to Reviewer ZaUC [2/2]**
>
> > **W2: It would be useful to know the best performance a single model could achieve with more advanced agent formulations.**
>
> Thank you for this insightful suggestion. While our primary results use MCPMark-Agent, a minimal tool-calling loop intended to expose models’ core MCP capabilities, we agree that it is valuable to explore more powerful scaffolds.
>
> We have conducted additional experiments using ReAct and Codex for representative models. These results will be included in the revision:
>
> | Model | Agent | Overall | Filesystem | GitHub | Notion | Playwright | PostgreSQL |
> |-------|--------|---------|-----------|--------|--------|------------|------------|
> | gpt-5-medium | MCPMark-Agent | 52.6% | 57.5% | 47.8% | 42.0% | 43.0% | 76.2% |
> | | ReAct | 37.8% | 49.2% | 39.1% | 21.4% | 19.0% | 64.3% |
> | | Codex | 36.2% | 33.3% | 30.4% | 25.0% | 20.0% | 81.0% |
> | gpt-5-mini-medium | MCPMark-Agent | 27.4% | 33.3% | 18.5% | 16.1% | 12.0% | 61.9% |
> | | ReAct | 26.6% | 35.0% | 22.8% | 14.3% | 15.0% | 48.8% |
> | grok-4-fast | MCPMark-Agent | 24.0% | 29.2% | 13.0% | 3.6% | 27.0% | 27.0% |
> | | ReAct | 26.0% | 31.7% | 21.7% | 28.6% | 11.0% | 36.9% |
> | glm-4.6 | MCPMark-Agent | 23.6% | 10.0% | 19.6% | 25.0% | 19.0% | 51.2% |
> | | ReAct | 22.1% | 10.8% | 21.7% | 25.0% | 14.0% | 44.0% |
>
> We find that the simple design (naive iterative tool-calling) provides a strong baseline, even outperforming scaffolds like Codex. We attribute this to the fact that sophisticated scaffolds are often unoptimized for multi-step workflows with MCPs and may introduce redundant context or constraints, particularly in the system prompt. While designing a custom scaffold for MCPMark could yield better performance, it is beyond the scope of this work.
>
> > **W3: Why not use MCP as a wrapper around existing multi-tool, multi-turn agent benchmarks instead of creating new ones?**
>
> Thank you for the question. Our core objective is to mimic the real-world challenges agents will face, which, as you suggest, requires benchmark tasks to be both highly diverse and realistic.
>
> Existing multi-tool or agent benchmarks, while valuable, often force a trade-off between these two properties. For example, SWE-Bench offers strong realism but is confined to the single domain of software engineering (e.g., GitHub issue debugging), limiting task diversity. Conversely, benchmarks like AppWorld provide task diversity across applications but often rely on custom-built wrappers or research-oriented APIs. This simulation cannot fully replicate the stateful, production-level challenges of live systems.
>
> This is the gap MCPMark addresses. By leveraging the Model Context Protocol (MCP), our benchmark uniquely combines official, production-grade MCP servers with real and diverse initial states. This high-fidelity setup is what allows us to test an agent's ability to handle the complex, stateful, real-world workflows they will encounter in deployment—a practical challenge that simulated benchmarks cannot replicate.
>
> Furthermore, leveraging a standard protocol broadens MCPMark's scope, enabling the evaluation of both agent models and MCP server implementations. This proves to be a critical variable, as performance varies significantly by server even when using the same model. For instance, with `claude-sonnet-4`, the KlavisAI GitHub server achieved a 31.5% pass@1 at a lower token cost (533,385 vs. 701,252), while the official GitHub server reached only 16.3% on the same tasks. Our detailed server implementation results are provided below:
>
> | Service  | MCP Server        | Model               | Pass@1 (avg ± std) | Avg. Tokens |
> |----------|--------------------|---------------------|---------------------|-------------|
> | GitHub   | KlavisAI          | claude-sonnet-4     | 31.5 ± 3.6%         | 533,385     |
> | GitHub   | GitHub Official   | claude-sonnet-4     | 16.3 ± 5.7%         | 701,252     |
> | Notion   | Klavis AI         | claude-sonnet-4     | 34.8 ± 6.4%         | 424,474     |
> | Notion   | Notion            | claude-sonnet-4     | 21.4 ± 5.1%         | 650,879     |
> | Postgres | InsForge          | claude-sonnet-4-5   | 54.8 ± 5.3%         | 391,019     |
> | Postgres | Supabase          | claude-sonnet-4-5   | 52.4 ± 5.8%         | 554,427     |
> | Postgres | Postgres Official | claude-sonnet-4-5   | 48.8 ± 4.0%         | 492,931     |
>
> Finally, we thank the reviewer for this constructive suggestion. We agree this is a great angle to make our work more complete and also add this discussion comparing MCPMark with existing multi-tool agent benchmarks to our manuscript in Section 2.2 (highlighted in blue)

---

### Author Response · Authors · 2025-11-27
**Summary of Paper Revision**

We thank all reviewers for their constructive feedback, and we have responded to each reviewer individually. We have also uploaded a **Paper Revision** including additional results, analyses, and clarifications:

* $\\textrm{\\color{blue}Sec. 2.2}$ (Page 4): We add a discussion on why MCPMark is necessary, including a comparison with existing tool and agent benchmarks.
* $\\textrm{\\color{blue}{Sec. 3.3}}$ (Page 7): We include evaluations of MCP servers and Agent scaffolds. The corresponding results are reported in $\textrm{\\color{blue}{Table 12}}$ (Page 43) and $\textrm{\\color{blue}{Table 3}}$ (Page 8).
* $\textrm{\\color{blue}{Table 13}}$ (Page 43): We add experiments comparing Codex and ReAct agent scaffolds under different reasoning effort settings.
* $\\textrm{\\color{blue}{Table 14}}$ (Page 43): We add results for the easy-subtask version of the benchmark, derived from the standard tasks by simplifying requirements or adding hints, with 10 tasks for each MCP server.

---

### Author Response · Authors · 2025-11-28
**Looking forward to further feedback**

Dear Reviewers,

Thank you again for your valuable comments and suggestions, which are really helpful for us. We have posted responses to the proposed concerns and included additional experiment results.

We totally understand that this is quite a busy period, so we deeply appreciate it if you could take some time to return further feedback on whether our responses solve your concerns. If there are any other comments, we will try our best to address them.

Best,

The Authors

---

### Meta-Review · Area_Chair_mso9 · 2026-01-09

**Summary:**

This submission proposes MCPMark, a comprehensive benchmark for stress-testing LLM agents’ realistic MCP use across 127 tasks. Reviewers raised concerns about task scalability, justification for 127 tasks, agent scaffold performance, comparison with existing benchmarks, and MCPMark-Agent details. The authors’ thorough rebuttal addressed the key concerns with additional experiments, clarifications, and supplementary materials.

**Reviewer Concerns:**

Addressed Concerns:

Reviewer ZaUC: Scalability (easy subset + community contributions), 127 tasks’ statistical adequacy/representativeness (detailed validation), advanced agent scaffold performance (ReAct/Codex experiments), MCP-specific benchmark necessity (comparison with AppWorld/SWEBench in Sec. 2.2).

Reviewer VEYJ: Data curation scalability (exploration-synthesis pathway), human expertise impact (domain-specific task examples), MCPMark-Agent scaffold (LiteLLM/MCP SDK details), scaffold effect (reasoning effort experiments), implicit errors (core model limits + supp examples), test-time scaling (evaluated; parallel rollouts discussed).

Reviewer NsiE: no concerns raised.

Outstanding Concerns: All reviewer questions and weaknesses were addressed.

**Reviewer Scores:**

Reviewer ZaUC (initial 6): Likely raise to 8 (all concerns resolved, additional experiments strengthen rigor).

Reviewer VEYJ (initial 6): Likely remains 6 (clarifications on scaffold, errors, and scalability address gaps).

Reviewer NsiE (initial 10): Remains 10 (the review is short and a little uninformative, no concerns raised).

---

### Decision · Program_Chairs · 2026-01-26

Accept (Poster)